# Analysis and Design of Typical Automated Container Terminals Layout Considering Carbon Emissions

**Nanxi Wang [1,\*], Daofang Chang [2], Xiaowei Shi [1], Jun Yuan [3] and Yinping Gao [2]**

[1] Logistics Engineering College, Shanghai Maritime University, 1550 Haigang Avenue, Pudong, Shanghai 201306, China; shixiaowei1990@hotmail.com

[2] Institute of Logistics Science & Engineering, Shanghai Maritime University, 1550 Haigang Avenue, Pudong, Shanghai 201306, China; dfchang@shmtu.edu.cn (D.C.); gaoyinping@stu.shmtu.edu.cn (Y.G.)

[3] China Institute of FTZ Supply Chain, Shanghai Maritime University, 1550 Haigang Avenue, Pudong, Shanghai 201306, China; yuanj@shmtu.edu.cn

\* Correspondence: 201830210253@stu.shmtu.edu.cn or 18328072968@163.com

**Abstract:** With the rapid development of world economy and trade and the continuous construction of green port, automated container terminal (ACT) has increasingly become the direction of future development. Layout design is the premise of ACT construction, which has an at least 50-year influence on the terminal. Therefore, this paper hopes to analyze and design the typical ACT layout to achieve sustainable development of the port. Firstly, a conceptual model is presented considering the interaction between different areas within the ACT when the width and length of the terminal are fixed. To select the optimal layout to achieve the goal of the green terminal, a novel mathematical model is established based on the energy consumption during cycle operation of various devices which can estimate the total carbon emission of an ACT over a period and is suitable for designing period. Then, with the developed model, an ACT in East China was taken as a case study. Finally, according to various analysis of the data results, the layout suggestion considering the sustainable development of the port is given.

**Keywords:** automated container terminals; layout; cyclic operation; carbon emissions; correlation analysis

## 1. Introduction

As the important hub for the realization of land and sea logistics transportation, container terminals play an extremely crucial role in the world's economic trade, and containerized trade accounts for 17.1% of total seaborne trade [1]. With the continuous development of the world economy and trade and the intensification of the competition of each wharf, wharf operators start to think about how to strengthen their own construction and achieve sustainable development from all aspects. Automated container terminals (ACTs) are not only the key development direction of the port in the future, but also the new revolution of port construction. Since 1993, when the world's first ACT, the ECT terminal at the port of Rotterdam, the Netherlands, was completed, the Port of Singapore, the Port of Hamburg, Thamesport of England and the TCB terminal of Nagoya Port in Japan have successively carried out the construction and commissioning of ACTs. According to the statistics of UNCTAD [1], more than 50 ACTs around the world have been built until 2017, because of their significant advantages in saving terminal manpower costs, improving port capacity, reducing equipment energy consumption, and enhancing the image of ports, and so on. Then, after more than 25 years of development and innovation, the current technology of ACTs has gradually matured and improved. With the progress of the development of science and technology, the demand of the development of the shipping market, and taking into account the rising costs of port enterprises, the frequent occurrence of safety accidents,

and personnel operation being unable to meet the requirements of development and other factors, an increasing number of traditional docks are considering, preparing or being built or transformed into ACTs. The investment in ACTs is large and the construction cost is high. Once completed, the cost of repair or reconstruction is great, too. For example, the cost estimate of The TraPac Terminal Program is 510,412,388 dollars in 2013 (The TraPac Terminal Program consists of 10 projects and will provide wharves, automated backlands, rail facilities, buildings, and gates for the Port of Los Angeles' first automated container terminal at Berths 136–147.) [2]. Therefore, it is of great importance to make a reasonable analysis of the layout of ACTs before construction or transformation.

The design of the ACTs was originally proposed by [3], who designed, analyzed and evaluated four different automated container terminals (ACTs) concepts, which included ACTs based on the use of automated guidance vehicles (AGVs), a linear motor conveyance system (LMCS), an overhead grid rail system (GR), and a high-rise automated storage and retrieval structure (AS/RS). Based on previous studies [4] and the current layout of ACTs in the world, of which process layout is shown in Table 1, this paper analyzes the most typical ACT layout with process mode of "Double trolley QC + AGV (Power) + ARMG(Stereo library ) ", which is schematically shown in Figure 1.

**Table 1.** Global automated container terminal process layout.

| Terminal Abbreviation | Process Mode | Shore Equipment | Level/Site Equipment | Passing Ability (TEU) | Production Time |
|---|---|---|---|---|---|
| Port of Hamburg HHLA CAT (The Second generation) | Double trolley QC + AGV + ARMG | 15 QCs | 86*AGV 52*ARMG | 3 million | 2002.06 |
| Port of Rotterdam Euromax (The Third generation) | Double trolley QC (2G) + AGV + ARMG | 16 QCs | 96*AGV 58*ARMG | 2.3 million | 2010.06 |
| Port of Xiamen Ocean Gate (The Fourth Generation) | Double trolley QC + AGV (power) + ARMG | 3 QCs | 18*AGV 16*ARMG | 0.91 million | 2016.03 |
| Port of Yangshan fourth phase (The Fourth Generation) | Double trolley QC + AGV (power) + ARMG (Stereo library ) | 16 QCs | 88*AGV 80*ARMG | 6.3 million | 2017.12 |

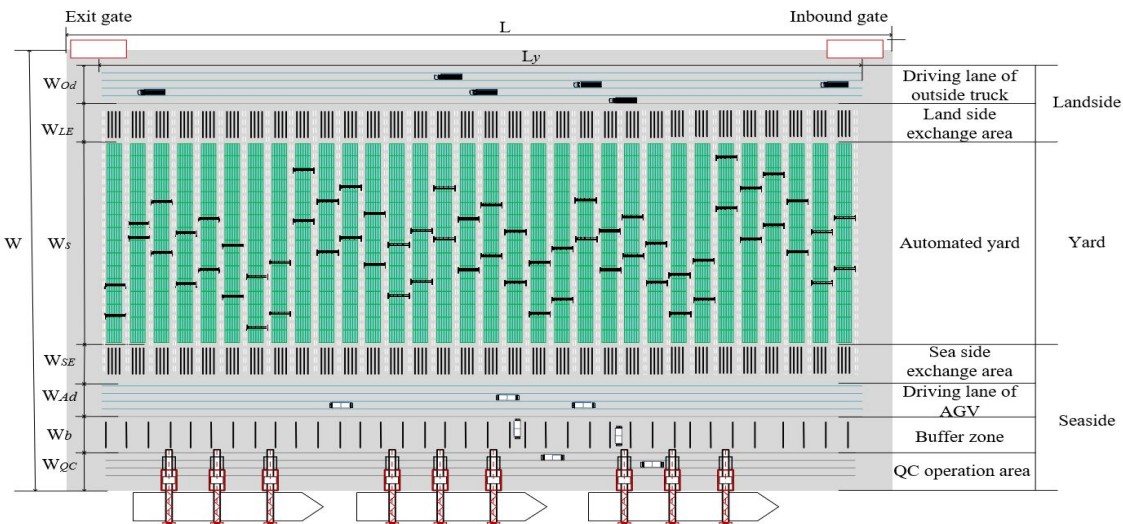

**Figure 1.** The schematic diagram of the layout of the typical automated container terminals (ACTs).

A typical ACT is divided into three parts: sea side, land side and yard. The sea side includes berth, quay crane (QC) operation area, buffer zone, AGV driving lane and the sea side exchange area, while the land side includes the driving lane of outside truck (OT), the land side exchange area

and gates to and from the port. The storage yard adopts vertical row block area. The blocks are laid out vertical with the gate or the berth, and the width of the block depends on the type of the selected Automatic Rail-Mounted Gantry Crane (ARMG) (the width of the block is less than the ARMG span used).

At present, considerable attention is being given to climate change and global warming. Global warming is the increase in the average temperature of the Earth's lower atmosphere air and oceans that has occurred since the mid-20th century and its projected continuation. As humans burn fossil fuels, such as petroleum, coal, etc., or cut down forests and burn them, they produce a large amount of greenhouse gases such as carbon dioxide ($CO_2$).

In response to climate change, governments have made various regulations and targets, the EU 2011 White Paper on Transport aims at a high-level target: reducing by year 2050 transportation-related GHG emissions by at least 60% with respect to 1990 levels. Other areas of the developed world (including North America, Japan, and Australia) have very similar goals for environmental improvement. Even in developing economies in Asia, South America and Africa, who believe that if they are subject to the same kinds of environmental guidelines as in developed economies may impede their own economic development, also take positive measures. For example, China has seriously been taking action on climate change for some years, with the publication in 2007 of China's first national action plan on climate change [5]. As seaports are important hubs and a major source of carbon emissions, reducing carbon emissions and energy consumption in the seaports is crucial to achieving the climate goal. Thus, China released the No. 315 document in 2011, which set a target of reducing carbon dioxide emissions per unit of port throughput by 10% [6].

In response to various policy provisions, terminal operators and shipping companies have also taken various measures, which can be classified into three levels.

- The technical level includes the use clean fuels, the upgrading of engines and the search for alternative energy, such as the use of cold ironing [7] and automatic mooring systems [8].
- The operational level includes the coordination of the operating mechanism of ships and trucks, for example terminal appointment system (TAS) [9], vessel-dependent time windows (VDTWs) [10] and limited entering time slots (LETS) [11].
- The economic level includes the change of pricing policies for tariffs and charges, such as tariff/toll pricing policies [12,13].

Nowadays, with the implementation of measures such as "oil to electricity" of terminal equipment, restriction of ship route and speed, emission reduction has achieved certain results. For example, as shown in Figure 2, carbon emissions from the Los Angeles port have been controlled over the past decade since the implementation of CCAP in 2005, although there has been slight fluctuations in carbon emissions in recent years as a result of the increase in the number of containers (The port of Los Angeles (POLA) and the port of Long Beach have established the San Pedro Bay Ports Clean Air Action Plan (CAAP) to reduce pollution from their production since 2005. As the biggest container terminal in the USA, the POLA has published a full report each year called "Air Emissions Inventory" [14] since the beginning of CAAP. Based on the data from the report, Figure 2 shows the trend in the $CO_2$ equivalent emissions over 2005–2017.). Despite the construction of green ports and consideration of future sustainable development, more efforts still need to be tried.

Considering the layout of the container terminals is one of the most influential factors in the productivity of the container handling operations, which significantly affects terminal performance under different transporter dispatching rules and allocation strategies [15]. This paper seeks to optimize the layout of ACTs with the lowest carbon emissions by establishing a carbon emission calculation model based on the energy consumption during cyclic operation of various devices in the ACT, which can provide advice for future ACTs construction.

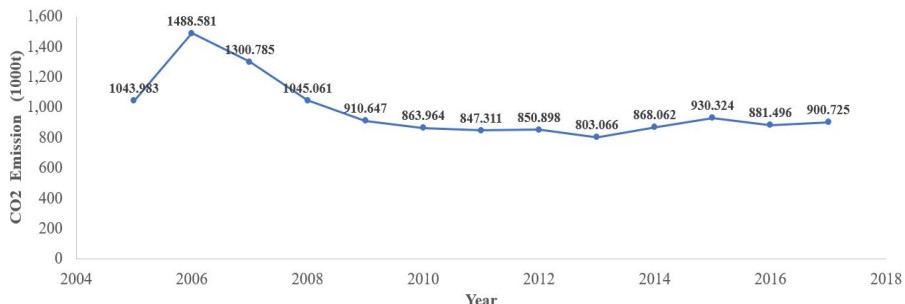

**Figure 2.** The total maritime industry-related $CO_2$ emissions in POLA from 2005 to 2017.

This paper is structured as follows. Section 2 provides a review of the existing literature on layout optimization of container terminals, carbon emission problem of ports and the methodology of carbon emission calculation. Section 3 describes the layout problem of a typical ACT and provides optimized layout model of ACTs. The cycle of various equipment during each operation is described in detail in Section 4 and the total carbon emissions calculation model is given. Then, a case study is presented in Section 5. Based on the case results, analysis from partial to overall and design suggestions considering the sustainable development of the terminal are given. Section 6 provides the conclusions and discussions of the proposed model.

## 2. Literature Review

### 2.1. Layout Optimization of Container Terminals

In the study of container terminals, there are few studies on layout design. The design of container layout mainly concentrates on the design of yard layout, and the design of yard layout mainly focuses on three parts: the allocation of resources, the selection of operation technology, and the optimization of the length or width of blocks. The technologies used are mainly the derivation of mathematical formulas and simulation experiments.

In terms of the mathematical formulas, considering the impact of the stack height and the number of layers in the block, Kim [16] first derived a simple formula to estimate the expected number of re-handles for a random pick-up in a given bay by regression analysis. Then, considering the expected number of relocations for picking up a container from a given layout of transfer cranes and the expected travel distance of yard trucks for delivering and picking-up in a given layout will vary with the layout of the yard, Kim et al. [17] came up with a method to estimate the impact of different variables on operating costs and used it to compare the layout in which blocks are laid out parallel to the berth with the layout that blocks are vertical. Wiese et al. [18] introduced different formulations for planning the yards of arbitrary shaped container terminals considering yard layout with transfer lanes. Gupta et al. [19] captured the stochasticity with an integrated queuing network modeling approach to analyze the performance of container terminals with parallel stack layout using automated lifting vehicles. After investigating 1008 parallel stack layout configurations on throughput times, they found that, assuming an identical width of the internal transport area, container terminals with parallel stack layout perform better (4–12% in terms of container throughput times) than terminals with a perpendicular stack layout.

To cope with the high uncertainty involved in the complex terminal system, a simulation model is always used to solve the yard layout problem [20]. Zhang et al. [21] used simulation technology to study the influence of length of container yard on terminal operation efficiency. Kemme [22] conducted a simulation study to evaluate the effects of four rail-mounted-gantry-crane systems and 385 yard block layouts—differing in block length, width, and height—on the yard and terminal performance. Petering [23] investigated how the width of the storage blocks in a terminal's container yard affects the overall, long-run performance of a seaport container terminal as measured in terms of GCR (average quay crane work rate) and they found that, to keep QCs busy and minimize the makespan of the

schedule of ships, the block length should be limited between 56 and 72 TEU. Considering the uncertain throughput in the future, Zhou et al. [24] proposed an optimization framework based on simulation to obtain a cost-effective and reliable design solution to the physical layout and equipment deployment strategy of the yard at a mega container terminal. In addition, the optimization and simulation research of container yard layout are also launched based on Flexsim [25], improved SLP theory [26].

Since throughput, yard capacity, equipment cycle time, etc. are important performance indicators of the terminal, the first two were taken by Lee and Kim [27] to optimize the block size by two methods. The expected cycle times of the straddle carriers (SCs) was presented by Wiese et al. [28] considering the parallel and perpendicular layout as well as various driving and storage space compensation strategies. Lee and Kim [29] proposed a method for determining an optimal layout of container yards taking into consideration the storage space requirements and throughput capacities of yard cranes and transporters. In recent years, more problems have been considered by researchers. Martin et al. [30] presented a method for forecasting the yard inventory of container terminals over an extended period by developing a formulation based on random variables and probabilistic functions, and addressed an integrated yard planning problem for determining the optimal storage space utilization by considering the yard congestion effect on terminal performance. Lee et al. [31] aimed to discuss a design process to maximize the throughput capacity, as well as minimize the resource configuration when designing the yard layout. They also found that the single-lane yard layout is preferable when high throughput capacity is required, whereas the double-lane yard layout is superior in favor of high efficiency of vehicle flows. Dkhil et al. [32] both integrated the straddle carriers scheduling problem and the location assignment problem, which insures higher theoretical optimality, and studied the integrated problem as a multi-objective problem by evaluating eight realistic objectives to optimize operating times, storage space organization and the number of required straddle carriers. Zhou [33] studied the optimization of land layout of foreign trade container terminals considering uncertainties. Until now, no articles combine layout of ACTs issues with carbon emissions.

Despite the rapid development of ACTs, the research on the layout of container terminals is mostly concentrated on traditional terminals, while automated terminals are less involved. Liu et al. [34] studied the effect of the yard layout on the performance of ACTs by considering two commonly used yard layouts and developing simulation models. For each automated yard layout, they considered and compared three operational scenarios: loading, unloading, and combined loading and unloading operations. Wang et al. [35] selected four representative overseas automated container terminal yard layout forms for analysis, and combined the characteristics of plane layout, equipment selection, functional planning and other aspects of reference. Then, combined with the actual situation of Yangshan Phase 4 ACT, they proposed a new model of automated yard layout. However, the analysis lacks a mathematical model.

### 2.2. Carbon Emission Problem of Ports

The carbon emission problem of ports has long been widely studied by experts in various fields, with research objects ranging from individual equipment such as ships, QCs, yard cranes, etc.; to part areas of the terminal such as yards, berths, etc.; and to the entire terminal.

In terms of the whole terminal, Geerlings and Duin [36] presented a methodology to analyze the $CO_2$-emissions from container terminals, illustrated by the Port of Rotterdam. They showed that changing the original straddle carriers (SCs) to electric straddle carriers ( ESCs) made it possible to reduce the $CO_2$-emissions of the current terminals by nearly 70%, although it was costly. Kim et al. [37] developed a multimodal container freight network design problem (NDP) to reduce GHG emissions below a target. By including emission cost in the objective function and the maximum emission constraint in the upper level problem, the NDP provided an optimal combination of investment alternatives to minimize the total system cost and to meet the emission reduction target. Yun et al. [38] established a carbon emission quantification simulation model considering four kinds of mitigation strategies as inputs: reduced speed in waterway channels, reduced auxiliary

time at berth, onshore power supply and alternative fuels, and increased working efficiency of port equipment.Yang and Lin [39] employed a green container terminal perspective to compare the performance of four types of cargo handling equipment used in container yards—automatic rail, rail, electric tire, and tire transtainers—based on working efficiency, energy saving performance, and carbon reductions, and found that automatic rail and electric tire transtainers were the optimal types of green cargo handling equipment.

In terms of the part area of the terminal, Yu et al. [40] built an assignment model of export container to capture the behavior of tractor arrivals at each block when the loading begins and used the queuing theory to model the congestion happening in the yard, and then evaluated emissions from yard tractors based on the forecast arrivals. Peng et al. [41] established a simulation model to quantify the impact of the allocation of facilities, including the number of facilities and the fuels adopted by facilities, on carbon emissions. Hu et al. [42] solved the berth and quay-crane allocation problem, which considers fuel consumption and emissions from vessels. Venturini et al. [43] introduced a novel mathematical formulation that extends the classical BAP to cover multiple ports in a shipping network under the assumption of strong cooperation between shipping lines and terminals.

In terms of the individual process equipment, Peng et al. [20] tried to solve the energy replacement problem (electric rubber tire container gantry cranes were used to replace rubber tire container gantry cranes) at a network level and coped with the high uncertainties in the container terminal transportation network. Therefore, they modeled the energy replacement problem with the purpose of minimizing the carbon emissions by combining an allocation resource mathematical model and a simulation model of the whole transportation network together. Minh and Huynh [44] applied the diffusion approximation M/G/n queuing model to provide an analytical tool for assessing and optimizing the gate layout for a given truck arrival rate and truck service rate under two gate queuing strategies: pooled and non-pooled. Schulte et al. [45] introduced a collaborative planning model to be operated within a truck appointment systems (TAS) and to investigate its impact on emission and cost objectives. Chen et al. [46] proposed a methodology to optimize truck arrival patterns to reduce emissions from idling truck engines at marine container terminals and developed a bi-objective model to minimize both truck waiting times and truck arrival pattern change. Zis et al. [47] extended existing literature to present a consistent and transferable methodology that examines emissions reduction port policies based on ship-call data. Do et al. [11] developed a method to optimize the time slot assignment for individual trucks, aiming at minimizing total emissions from trucks and cranes at import yards. Díaz-Ruiz-Navamuel et al. [8] aimed to verify the effect of the Automatic Mooring Systems (AMS) on the emission of pollutant gases in the surroundings of the installations devoted to Ro-Ro/Pax vessel traffic which focusing on the $CO_2$ emissions produced by vessels during mooring operations using two different calculation methodologies (EPA and ENTEC). Then, Ortega Piris et al. [48] studied the reduction in the $CO_2$ emissions of merchant vessels as a consequence of the substitution of traditional mooring systems with the new automatic systems presented for the first time by Díaz-Ruiz-Navamuel et al. [8].

To the authors' knowledge, no article considers reducing the carbon emissions of the entire terminal by optimizing the layout of the automated container terminals, including planning the detailed width of every part of ACTs. Therefore, this paper is a first attempt to obtain environmental footprints from terminal layouts.

*2.3. The Methodology of Carbon Emission Calculation*

In general, there are two main methods that can be used to produce fuel consumption and emission estimation for transportation activities. The first method is called the "top-down" method, or "fuel-based" method, and uses fuel sales to estimate emissions. This would be the most reliable method of estimating total fuel consumption and emissions if the figures of fuels sales reported were absolutely reliable. Since fuel consumption data are sometimes not available or unreliable, the so-called "activity based", or "bottom-up", method is used more widely. In activity-based approaches, one can

estimate emissions based on modeling of the transportation activity or by using conversion factors that convert the available data into emissions.

Combining different application scenarios and diverse research methods, carbon emissions are calculated in a variety of ways.

Yang [49] used carbon footprint analysis and gray relational analysis to investigate $CO_2$ emissions produced by two different container terminal operating models (tire transtainers and rail transtainers) at the port of Kaohsiung, and sought to determine energy saving and $CO_2$ reduction strategies for shipping companies and terminal operators to comply with green port requirements. An activity-based emissions model was used to estimate the $CO_2$ emissions of container transport under four scenarios where there are switches of market share from existing ports to the emerging port and the results showed that there are greater reductions in $CO_2$ when transshipment routes are changed from the ports of Kaohsiung, Taichung and Keelung to the emerging port of Taipei [50]. Sim [51] proposed a model using a system dynamics approach to evaluate the total amount of carbon emissions produced in a container terminal, and calculated the required reduction amount of carbon emissions in the container terminal at a given carbon emission reduction goal from 2017 to 2030. The results of this study indicate that the container terminal will produce annually on average 108.18 million kg of $CO_2$ equivalent emissions from the five types of processes (vessel maneuver, vessel at berth, container loading and unloading, container transportation, and container receiving and delivery) from 2017 to 2030. Johansson et al. [52] evaluated the emissions of $SO_x$, $NO_x$, $CO_2$, CO and $PM_{2.5}$ using the Ship Traffic Emission Assessment Model (STEAM) by combining the information on individual vessel characteristics and position reports generated by the automatic identification system (AIS).

Different methods are suitable for different scenarios, considering that various operational data are difficult to obtain at the design stage, and throughput and device type are the key points to be considered at the time of design. Therefore, the carbon emission calculation model based on the number of containers and equipment design operation parameters can be established to estimate the carbon emissions of the future container, which is introduced in detail in Section 4.

## 3. Problem Description

### 3.1. Layout Description of ACT

As shown in Figure 1, the shape of a typical ACT can be regard as a rectangle. Before designing the layout of the terminal, the total length and width of the terminal are known. According to the investment amount, the type and quantity of various equipment selected can be determined. Therefore, this article discusses the impact of ACTs layout on carbon emissions, assuming the length and width of ACTs, and the number and type of operating equipment are known.

Assuming that the safe distance between each part area is included in the area, we can derive the following formula, which is also the constraint when designing the layout.

$$W = W_{od} + W_{LE} + W_{SE} + W_{Ad} + W_b + W_{QC} + W_s \tag{1}$$

where $W_{Od}$ is the width of the OT driving lane; $W_{LE}$ is the width of the land side exchange area (LE); $W_{SE}$ is the width of the sea side exchange area (SE); $W_{Ad}$ is the width of the AGV driving lane; $W_b$ is the width of the buffer zone; $W_{QC}$ is the width of the QC operation area; $W_s$ is the width of the storage yard (SY); and $W$ is the total width of the ACT.

For some areas, once the device specification is selected, the width of the area can be more or less determined. Taking the LE as an example, the layout diagram of LE is shown in Figure 3.

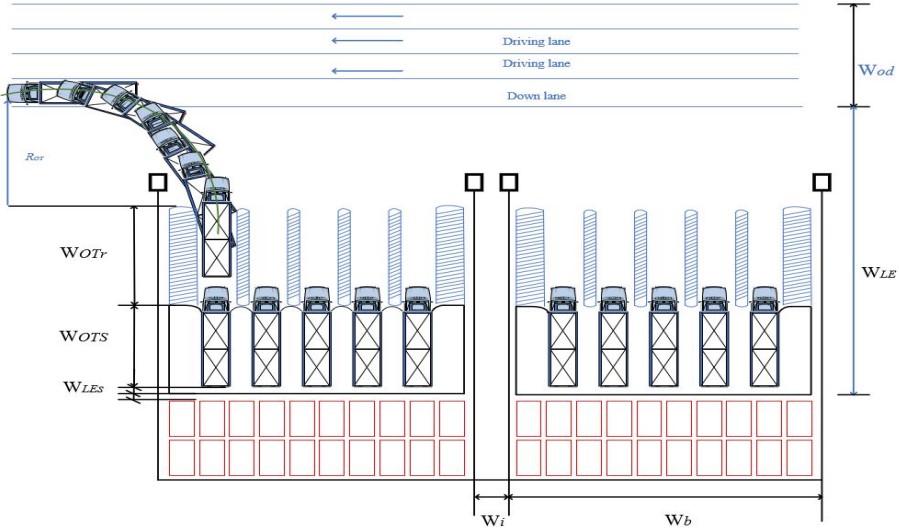

**Figure 3.** The layout diagram of the land side exchange (LE) area.

The OTs enter the front of the loading and unloading slot of LE by turning and reversing on the reverse lane of the outer truck lane, and then directly reversing into the designated loading and unloading area. The entire LE consists of the turning area of the OTs, the direct reversing area, the loading and unloading slot and the safe area. Therefore, the following formula can be derived.

$$W_{LE} = R_{OT} + L_{OTr} + L_{OTs} + L_{LEs} \qquad (2)$$

where $R_{OT}$ represents the turning radius of the OTs, which depends on the tonnage of the trucks; $L_{OTr}$ is the distance of OT direct reversing, which is equal to the length of OTs plus a fixed value of safety distance; $L_{OTs}$ is the length of OT slots in the loading and unloading area, which is equal to the length of OTs plus a fixed value of safety distance; and $L_{LEs}$ is the safe distance of the land side exchange area, which is a fixed value.

Therefore, once the type of the OT is determined, the above parameters can be determined, and thus the width of the land side exchange area can also be determined.

Similarly, the layout diagram of the sea side exchange area is shown in Figure 4, and the formal calculation is shown as follows.

$$W_{SE} = R_{AGV} + L_{AGVr} + L_{AM} + L_{SEs} \qquad (3)$$

where $R_{AGV}$ represents the turning radius of the AGVs, which depends on the tonnage of the AGVs; $L_{AGVr}$ is the distance of AGV direct reversing, which is equal to the length of AGVs plus a fixed value of safety distance; $L_{AM}$ is the length of AGV mates, which is equal to the length of AGVs plus a fixed value of safety distance; and $L_{SEs}$ is the safe distance of the sea side exchange area, which is a fixed value. Once the type of AGV is determined, the width of the sea side exchange area is set.

As for the buffer zone and QC operation area, we can see their diagram in Figure 5. The width of QC operation area is equal to the length of QC on the landside, which is equal to the length of the rail gauge plus the length of the rear beam of QC and the safety distance between QCs and the sea, and the width of buffer zone is equal to the length of an AGV plus twice the turning radius of AGV.

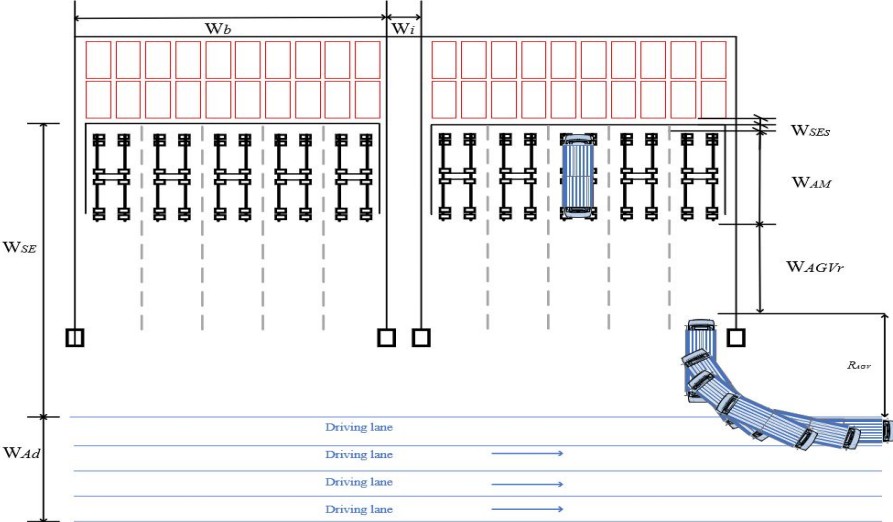

**Figure 4.** The layout diagram of the sand side exchange (SE) area.

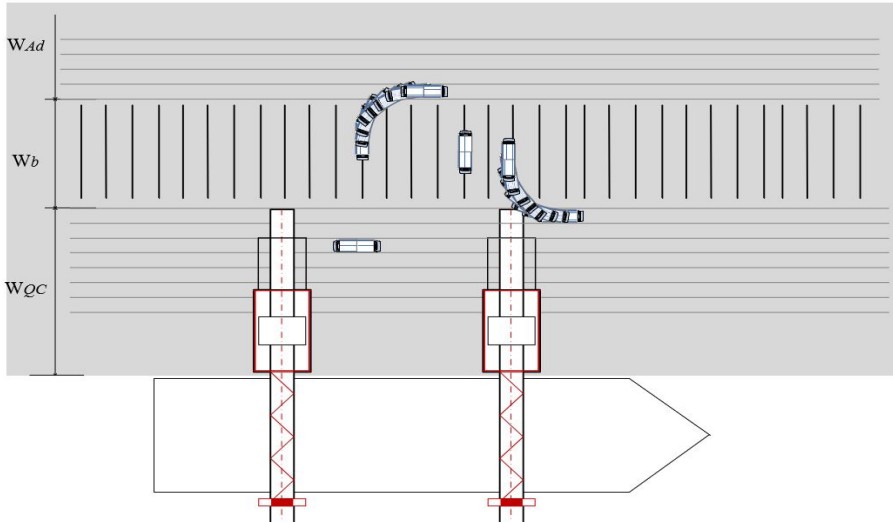

**Figure 5.** The layout diagram of the buffer zone and QC operation area.

In general, once the operating equipment is selected, the width of part of the ACTs (such as the land/sea side transfer area, the buffer zone and QC operation area) can be determined. Therefore, when designing the ACT, the width of the above four areas can be assumed to be constant. The remaining areas (the storage yard and the OT/AGV driving lanes) must be determined by decision makers based on other factors, such as throughout, working efficiency. In this paper, we determine the optimal layout based on the object to obtain least carbon emissions, while carbon emissions are positively related to the use of fuel, which is the layout with the least amount of energy.

In terms of the driving lane of OTs, the layout diagram of the driving lane of OTs can be seen in Figure 6. $n_{Od}$ is the number of lanes of the OT driving lane ($n_{Od}$ is a positive integer). Then,

$$W_{Od} = (n_{Od} - 1) * w_{Od} + w_{Odd} \tag{4}$$

$$w_{Od} = w_{OT} + w_s \tag{5}$$

where $w_{Od}$ is the width of a driving lane of OTs, $w_{Odd}$ is the width of the down lane, $w_{OT}$ is the width of the OTs and $w_s$ is the safe distance between driving lanes. From an economic point of view, $w_s$ should be less than $w_{OT}$. Thus, the value of the safe distance ranges from 0 to $w_{OT}$, which can be seen

as a fixed value. As for $w_{Odd}$, it should bigger than the $w_{Od}$ as the OTs are going to perform reverse operations on the down lane, while, from an economic point of view, $w_{Odd}$ should be less than $2w_{Od}$.

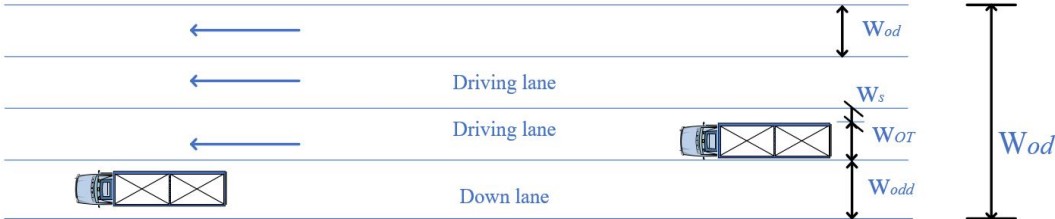

**Figure 6.** The layout diagram of the driving lane of OTs.

The AGV does not distinguish between the front and rear of the car, thus there is no special down lane. In practice, the width of a single driving lane of AGV $w_{Ad}$ can be set as a fixed value. If the number of driving lane of AGVs $w_{Ad}$ is set, the width of the driving lane of AGVs can be calculated as follows.

$$w_{Ad} = n_{Ad} * w_{Ad} \tag{6}$$

For a typical ACT, the layout diagram of the SY is shown in Figure 7. Except for the area of storing containers, it also includes the safety area to the LE and the SE, and the safety distance to LE or SE can be seen as a fixed value. Thus, the formal calculation of the SY is as follows.

$$W_S = L_b + 2w_{ss} \tag{7}$$

where $L_b$ represents the length of the blocks and $w_{ss}$ represents the length of the safety area from the storage yard to adjacent area.

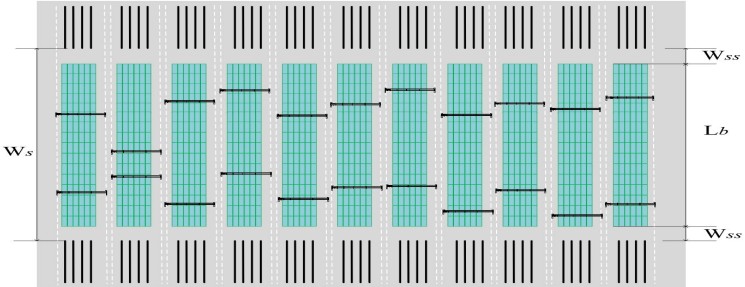

**Figure 7.** The layout diagram of the storage yard.

Before introducing the calculation of the block, the configuration of the container stacking is illustrated. As shown in Figure 8, a block can be represented by several bays, and there is a small gap between the bays to facilitate trolley extraction. A bay can be regarded as a stack of several rows plus several tiers of containers, which are the width and height of the bay. In general, an automated block has 10 rows and a maximum of six tiers, which depends on the type of the ARMG.

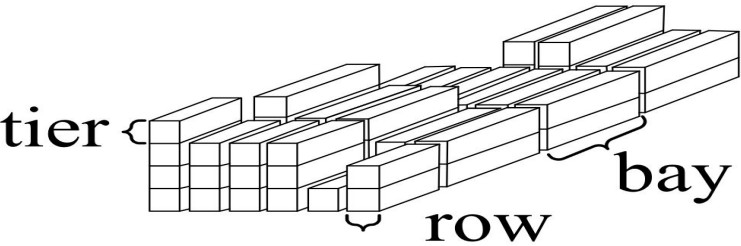

**Figure 8.** The configuration of a container stacking block.

In the ACT, one to three ARMGs are generally responsible for loading and unloading containers in a block, as shown in Figure 9. The width of the block area is smaller than the span of the equipment, and the height of the block area is lower than the height of the equipment. There is an interval between the containers, generally set to 0.3 m. Thus, the length of the block can be calculated as follows.

$$L_b = n_b * l_c + (n_b - 1) * 0.3 \tag{8}$$

where $n_b$ is the number of bays and $l_c$ is a 20-foot container length, equal to 6.1 m.

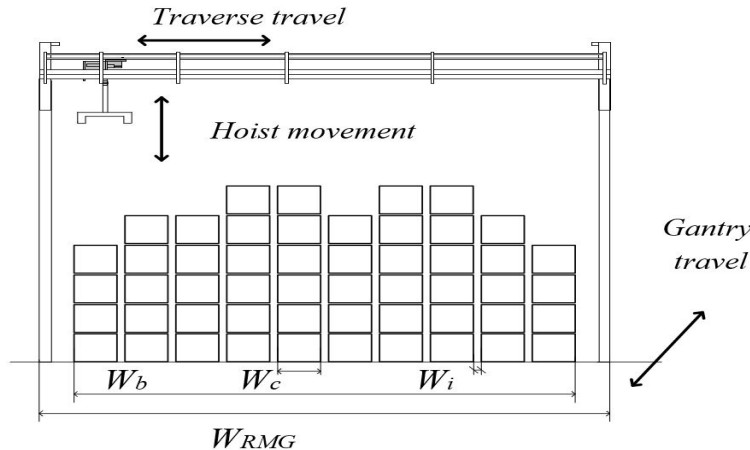

**Figure 9.** The Side view of an automated block.

After analysis, we find that $W_{LE}$, $W_{SE}$, $W_b$ and $W_{QC}$ can be regarded as constant, $L_b$ is related to $n_b$, $W_{Ad}$ is related to $n_{Ad}$, and $W_{Od}$ is related to $n_{Od}$. The number of bays affects the storage of the terminal, which affects the operation of the equipment and the operational efficiency of the entire terminal. The number of OT/AGV driving lanes affects the circulation path of the OTs/AGVs, which in turn affects the operational efficiency of the terminal. In the case of throughput determination, if the length of the block is too short and the driving lanes are too many, the average number of stacks of containers will increase, which is not conducive to the operation of the ARMG. If the size of the block is designed to be long and the number of driving lanes is reduced, it may cause the vehicles to be crowded and reduce the efficiency, thus it is necessary to choose a suitable layout size to achieve an optimal effect in the later operations.

In summary, the problem discussed in this paper is as shown in Figure 10. The width of the entire ACT can be regarded as a circle with a certain circumference. Except for some fixed length area such as the LE, three areas (SY, Od, and Ad) can be changed. Each variable area has the smallest distance value under certain constraints (this paper has a certain throughput). As the distance of each variable area increases, it will cause the motion of the equipment in the area to change and affect other areas; for example, when the distance of the SY increases, the average driving distance of the ARMG's gantry will increase, while the average stacking height of containers is reduced and the number of re-handles is reduced, which causes the movement of the ARMG's trolley to be reduced, and other areas also to be reduced.

Therefore, in this paper, $n_b$, $n_{Ad}$, and $n_{Od}$ are used as decision variables, and the optimal quantity combination is discussed to minimize the carbon emissions generated by equipment during loading and unloading of containers in operation.

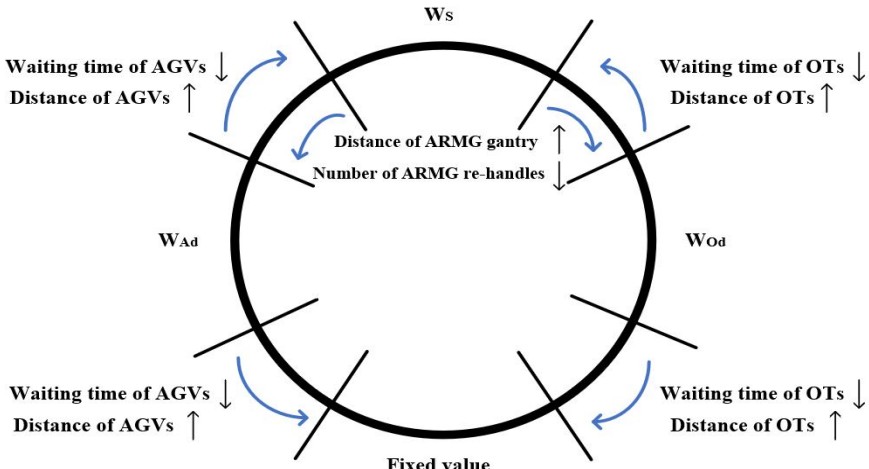

**Figure 10.** The conceptual diagram of layout problems in ACTs.

### 3.2. Design Constraints

When designing the number of lanes and bays in the block, we need to consider the requirements of traffic flow, throughput, etc. These constraints are discussed in the following sections.

To meet the traffic demand of the inland transportation of the terminal, when designing the number of driving lanes of the OTs or AGVs, road density should be considered, which needs to be limited. This article does not discuss in detail how to calculate road density, according to the "General Harbor Design Code" (JTS 165-2013) of the Ministry of Transport (China). The number of lanes required for the container terminal gate can be calculated as follows.

$$N_d = \frac{Q_h(1 - K_b)K_{BV}}{T_{yk}T_d p_d q_c} \tag{9}$$

where $N_d$ is the number of lanes required for the container terminal gate; $Q_h$ is annual throughput of the container terminal (TEU); $K_b$ is the percentage of the total container volume of railway transit, dismantling and water transfer containers within the inland area of the container terminal gate accounts for the annual volume of the terminal (%); $K_{BV}$ is the imbalance coefficient of container vehicles arriving at the port, 1.5–3.0; $T_{yk}$ is the number of working days (d) of the container yard, 350–365 d; $T_d$ is the working time of the gate (h), 12–24 h; $p_d$ is the number of vehicles passing through a single lane (car/h), 20–60 vehicles/h; and $q_c$ is the average vehicle capacity (TEU/vehicle), 1.2–1.6 TEU/car.

Similar to the calculation of the number of gates at the terminal gate, the number of driving lanes at the front of the terminal can be calculated as follows

$$N'_d = \frac{Q_h K'_{BV}}{T_{yk}T'_d p_d q_c} \tag{10}$$

where $N'_d$ is the number of lanes required for the front of the terminal; $K'_{BV}$ is the imbalance coefficient of container vehicles arriving the front of the terminal (in the automated terminal, the horizontal transportation equipment is generally scheduled by the central control, thus the imbalance coefficient is relatively low), 1.1–1.3; and $T'_d$ is the working time of the front of the terminal (h) (in ACTs, the horizontal transportation equipment works with QCs, thus their average working time is generally the average working time of the QCs), 12–24 h.

To meet the traffic demand of the inland transportation of the terminal, when designing the number of driving lanes of the OTs or AGVs, the constraint of the above formula should be satisfied.

At present, the stacking height of the container is up to six layers, thus the maximum number of layers set in this paper is also six. In the case of a certain throughput, the ground slots need to meet a certain amount to ensure that the container storage height does not exceed six.

## 4. Carbon Emission Calculation Model

As Geerlings and Duin [36] stated in their work, the $CO_2$ emissions are a direct consequence of the burning of fossil fuels to generate the energy needed to operate terminal processes. The transshipment of containers takes place with the different types of equipment that are used by the terminals. The type of equipment and the use of this equipment determines the energy consumption, and consequently the amount of $CO_2$ emissions. Since $CO_2$ emissions are the direct consequence of energy used by the transshipment process, it is important to obtain an idea of the factors in the transshipment processes that consume energy. These factors include the equipment used by each sub-process, the energy-consumption pattern of various types of equipment, the deployment of the equipment in each sub-process, and the average distance within a sub-process.

Therefore, when building the calculation model, this paper classifies the operating equipment and the operation task first, which can be seen in Table 2. As introduced in the Section 1, the main equipment types in the typical ACTs analyzed in this paper are ARMGs, AGVs, and QCs as well as the arriving vessels and OTs (Generally, ARMGs, AGVs, QCs use electricity as energy, while OTs use diesel, vessels use shore power after berthing, and use diesel when sailing). For ACTs, there are mainly four types of operation tasks, namely delivering, loading, discharging and picking-up, which are described in detail below. Then, the average energy consumption of each device when performing each task is calculated. Assuming that each device only operates with one container (20 ft/40 ft/45 ft) for each task, then the total energy consumption is related to the number of containers, i.e., the average energy consumption multiplied by the number of containers participating in the task, and accumulated finally to get a total carbon emissions calculation model of ACT. Compared with the previous research, a more detailed and complete carbon emission calculation model is established in this paper.

**Table 2.** Types of equipment and operation task at a terminal.

| i | i (Equipment) | j | j (Operation Task) |
|---|---|---|---|
| 1 | OT | 1 | delivering |
| 2 | ARMG | 2 | loading |
| 3 | AGV | 3 | discharging |
| 4 | QC | 4 | picking-up |
| 5 | Vessel | | |

All devices cooperate with other devices to participate in several or all of the operations. In the terminal, the equipment operation processes are all quite different. In the following, we carefully analyze the operation flow of each equipment under each operation task to obtain the average carbon emission calculation model of each equipment under each task.

### 4.1. Energy Consumption Formulation of OTs

The OT is mainly used to transport containers from the terminal to the inland, or to accumulate inland containers to the terminal. The process for sending or taking containers at ACTs of OTs is shown in Figure 11.

In this process, the movement of the OTs can be divided into three categories, namely, unload driving, load driving and idling. When the truck is empty to pick up the container, or when it has finished delivering the container to get out of the terminal, it belongs to the process of unload driving. When the OT comes to deliver the container or pick-up the container from the terminal, the process is called load driving. When the truck is in a stop-and-go state due to congestion or other reasons, and when it is idle during ARMG operation, it is called idle driving. Thus, the model of OTs can be seen below.

$$D_{OT} = (\frac{l_{Ol}}{v_{Ol}} * P_{Ol} + \frac{l_{Ou}}{v_{Ou}} * P_{Ou} + t_{Oi}*P_{Oi}) * R_d/\rho_d \quad (11)$$

where $l_{Ol}/l_{Ou}$ is the distance traveled by a loading/unloading OT to deliver the container at the terminal; $v_{Ol}/v_{Ou}$ is the average speed when the OT is loaded/unloaded; $t_{oi}$ is the total waiting time when an OT is idling at the terminal; $P_{Ol}/P_{Ou}$ is quota power of the OT in the process of traveling with/without load; $P_{Oi}$ is quota power of the OT in the process of idling; $R_{OT}$ is the diesel consumption rate of OTs, 0.2 kg/kWh; and $\rho_d$ is the density of diesel.

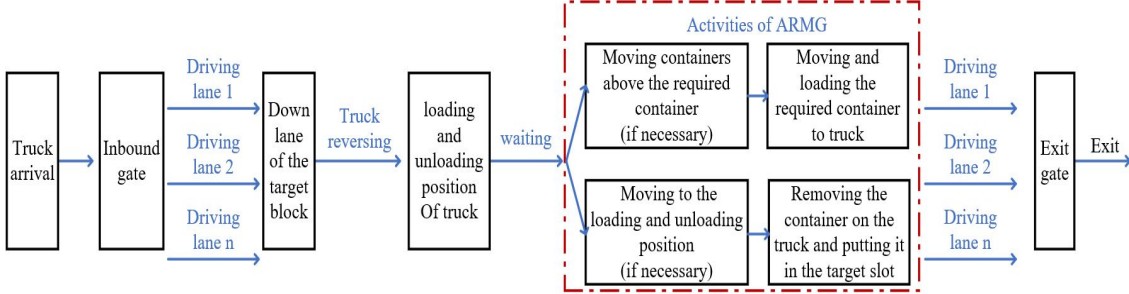

**Figure 11.** Process for sending or taking containers of OTs at ACTs.

### 4.2. Energy Consumption Formulation of ARMGs

As shown in Figure 9, the movement of ARMGs is divided into gantry movement and trolley movement, while the gantry is responsible for its forward and backward movement on the rail and the trolley is responsible for the traverse movement and the up and down movement when lifting the container or spreader. In the container terminal, ARMGs are responsible for the loading and unloading of the container in the yard, the operation of which can be regarded as a cycle. Taking the picking-up operation as an example, in the case of not considering the re-handles, the working cycle diagram is shown in Figure 12. Assume that the default position of the ARMG at the beginning of the cycle is with the spreader at its highest position. Other operations are similar only for the gantry's travel destination.

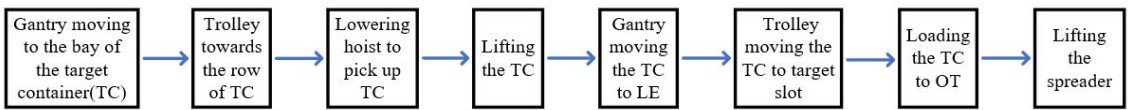

**Figure 12.** Working cycle diagram of ARMGs without re-handling.

When modeling AMRGs, the operation flow of an ARMG can be divided into six categories, namely, the operation of the gantry, the idle of the gantry, the traverse travel of the trolley without load, the up and down operation (hoist movement) of the trolley without load, the traverse travel of the trolley with load, and the up and down operation (hoist movement) of the trolley with load. Thus, the model of ARMGs can be seen below.

$$E_{ARMG} = \frac{d_g^m}{v_g^m} * P_g^m + \frac{d_t^{tu}}{v_t^{tu}} * P_t^{tu} + \frac{d_t^{hu}}{v_t^{hu}} * P_t^{hu} + \frac{d_t^{tl}}{v_t^{tl}} * P_t^{tl} + \frac{d_t^{hl}}{v_t^{hl}} * P_t^{hl} + t_g^i * P_g^i \quad (12)$$

where $d_g^m$ is average moving distance of the ARMG's gantry in the process of gantry operation; $v_g^m$ is average speed of the ARMG's gantry in the process of gantry operation; $P_g^m$ is quota power of the ARMG in the process of gantry operation; $t_g^i$ is average waiting time of the ARMG in the process of gantry idle; $P_g^i$ is quota power of the ARMG in the process of gantry idle; $d_t^{tu}/d_t^{tl}$ is average moving distance of the ARMG'S trolley during traverse traveling without/with load; $v_t^{tu}/v_t^{tl}$ is average speed of the ARMG's trolley in the process of the traverse travel without/with load; $P_t^{tu}/P_t^{tl}$ is quota power

of the ARMG in the process of the traverse travel without/with load; $d_t^{hu}/d_t^{hl}$ is average moving distance of the ARMG's trolley during hoist moving without/with load; $v_t^{hu}/v_t^{hl}$ is average speed of the trolley of ARMG in the process of the hoist movement without/with load; and $P_t^{hu}/P_t^{hl}$ is quota power of the ARMG in the process of the hoist movement without/with load.

This paper assumes that the operation of ARMGs is a continuous process, thus the process of gantry idle is not considered. In one cycle, the running distance of the gantry will vary with the length of the block and the cooperation with other ARMGs. The traverse travel of the trolley can be simplified to half of the span of the ARMGs. The up and down operation can be simplified to half of the lifting height of the trolley. In one cycle, the trolley of ARMG will do two up and down movements with/without load, and one transverse movement with/without load, thus the formula can be simplified to the following.

$$\overline{E_{ARMG}^c} = \frac{d_g^m}{v_g^m} + \frac{1}{2}d_s\left(\frac{P_t^{tu}}{v_t^{tu}} + \frac{P_t^{tl}}{v_t^{tl}}\right) + d_h\left(\frac{P_t^{tu}}{v_t^{tu}} + \frac{P_t^{tl}}{v_t^{tl}}\right) \tag{13}$$

where $\overline{E_{ARMG}^c}$ is the average power consumption in kWh of the ARMG in one cycle without considering re-handles; $d_s$ is the span of the ARMGs; and $d_h$ is the lifting height of the trolley.

However, in the actual container-intensive block, ARMG is not able to extract the target container directly, but needs to go through a more operation called re-handle.

Figure 13 shows a schematic diagram of a re-handle operation. To take out the target container, the other containers on the target container need to be moved to other places, and the process of re-handling one container can be simplified to two up and down movements without load, two up and down movements with load, one transverse movement without load, and one transverse movement with load, thus the average energy consumption formula for a re-handle operation can be as follows.

$$\overline{E_{ARMG}^r} = \frac{1}{2}d_s\left(\frac{P_t^{tu}}{v_t^{tu}} + \frac{P_t^{tl}}{v_t^{tl}}\right) + d_h\left(\frac{P_t^{tu}}{v_t^{tu}} + \frac{P_t^{tl}}{v_t^{tl}}\right) \tag{14}$$

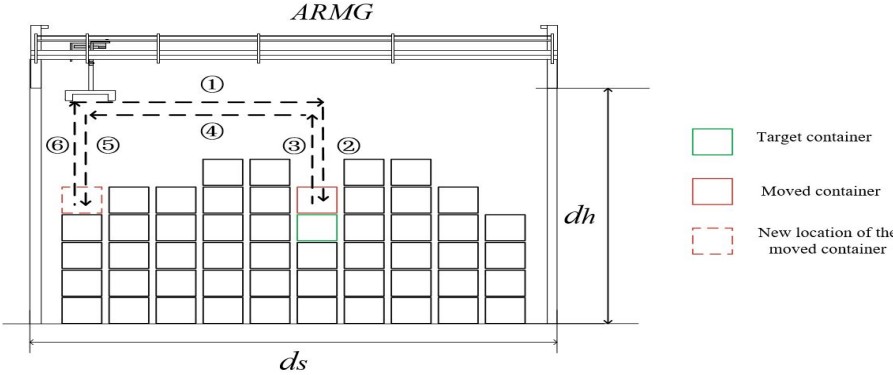

**Figure 13.** Working cycle diagram of ARMGs without re-handling.

In actual operation, there may be more than one container on the target container, and the stacking operation stipulates that the difference between the stacking layers of the adjacent two rows of containers shall not exceed two layers, thus, to obtain the target container, multiple re-handle operations will be performed. To approximate the expected number of re-handles for picking up an arbitrary container out of a bay, we can use the formula of Kim [16].

$$N_r = \frac{2\bar{h} - 1}{4} + (\bar{h} + 1)/(8a) \tag{15}$$

where $\bar{h}$ is the average stacking height and $a$ is the number of rows of the block (this formula assumes that every re-handled container is moved to a different slot in the same bay) [16].

According to the "General Harbor Design Code" (JTS 165-2013) of the Ministry of Transport (China), the number of ground slots required by the container terminal yard and the passing capacity are calculated as follows:

$$N_s = \frac{Q_h t_{dc} K_{BK}}{T_{yk} N_1 A_s} \tag{16}$$

where $N_s$ is the number of ground slots required by the container terminal yard (TEU); $Q_h$ is annual throughput of the container terminal; $t_{dc}$ is the average stockpile period of arriving containers (d); $K_{BK}$ is the container imbalance factor of the yard, 1.1–1.3; $T_{yk}$ is the number of working days (d) of the container yard, 350–365 d; $N_1$ is the number of stacking layers of the yard equipment; and $A_s$ is yard capacity utilization (%).

As discussed above, once the ARMG device type is selected, the number of rows in the block can be determined. The number of bays in the block is treated as a variable and the annual throughput and yard capacity utilization are treated as known data, thus the stacking height of the container $N_1$ can be determined according to above formula. The average stocking height can then be calculated as

$$\bar{h} = N_1 * A_s \tag{17}$$

Then, the expected number of re-handles $N_r$ can be determined. Therefore, when considering the re-handle operation, the energy consumption in the ARMG one cycle can be expressed as follows.

$$\overline{E_{ARMG}} = N_r * \overline{E^r_{ARMG}} + \overline{E^c_{ARMG}} \tag{18}$$

### 4.3. Energy Consumption Formulation of AGVs

AGVs are a horizontal transport tool used to assist QCs loading and unloading containers, which are only involved in the operations of loading and discharging. The transport path is shown by the dotted line in Figure 14.

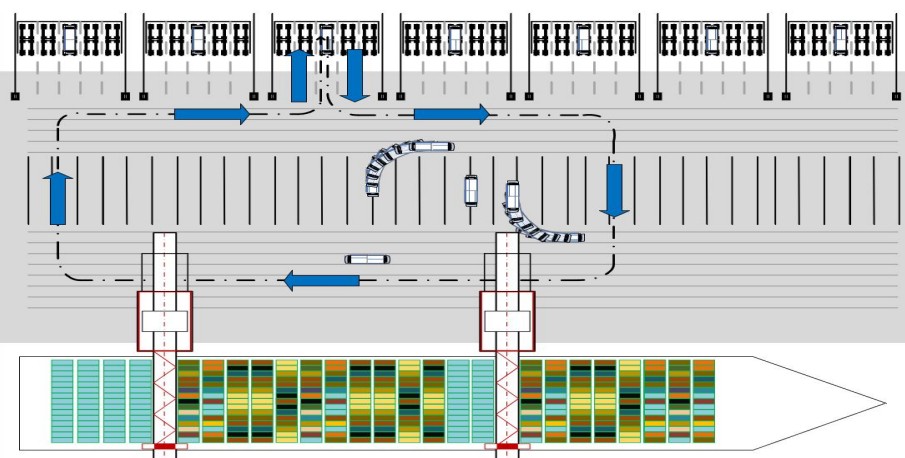

**Figure 14.** The diagram of AGV path.

During loading process, AGVs will arrive at the sea side exchange area to load the containers from the AGV mate or ARMG, transport them to the bottom of QCs in clockwise motion, and wait for QCs to pick up the container. After that, the AGVs return to the yard and transport the next containers. The discharging process is reversed. An AGV generally only serves one block or several adjacent

blocks, thus the entire motion of the AGV can be seen as a circular motion on the rectangle. The path that is transported once is the circumference of the rectangle, which can be expressed as:

$$C = (L_t + \frac{1}{2} W_{QC} + W_b + W_{Ad} + W_{SE}) * 2 \tag{19}$$

where $L_t$ is the distance of AGV horizontal transportation, which depends on the span of the AGV service blocks and the operating distance between the QCs serving the same vessel. Sometimes the AGV needs to serve multiple blocks, which can be regarded as the sum of the widths of these blocks. Although the AGV only serves one block, when the QCs operations are intensive and the working distance is not long enough, AGV is not convenient to turn between adjacent QCs, and also needs to travel across multiple blocks.

Similar to the OTs, the AGVs operation process can also be divided into three parts: unload driving, load driving and idling. Thus, the average power consumption of the AGVs for transporting a container can be expressed as follows.

$$E_{AGV} = \frac{d_{Al}}{v_{Al}} * P_{Al} + \frac{d_{Au}}{v_{Au}} * P_{Au} + t_{Ai} * P_{Ai} \tag{20}$$

where $d_{Al}/d_{Au}$ is the distance traveled by an AGV in the process of traveling with/without load; $v_{Al}/v_{Au}$ is the average speed when the AGV in the process of traveling with/without load; $P_{Al}/P_{Au}$ is quota power of the AGV in the process of traveling with/without load; $t_{Ai}$ is the total waiting time when an AGV is idling at the terminal(min); and $P_{Ai}$ is quota power of the AGV in the process of idling.

The specific process of loading and unloading containers by AGVs is shown in the Figures 15 and 16, which means AGV does not need to stay too much in the exchange area because of the help of the AGV mate. With the help of the buffer zone, there is no need to consider the congestion problem during the operation of the AGV. When the congestion is predicted by the system, the AGVs will be scheduled to wait in advance in the buffer zone. When the AGV is suspended, it only needs to consume a small amount of electricity, which can be neglected, thus this paper does not consider the problem of AGV idling. Therefore, it is only necessary to assume that the process of AGV transporting a container on average can be divided into two stages: empty and load.

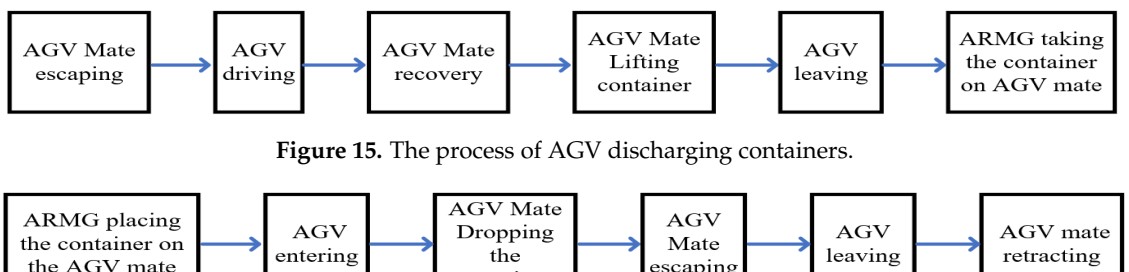

**Figure 15.** The process of AGV discharging containers.

**Figure 16.** The process of AGV loading containers.

### 4.4. Energy Consumption Formulation of QCs

QCs are the key equipment for connecting terminals and vessels, which are involved in the operations of loading and discharging. Double trolley QCs are generally used in ACTs. The schematic diagram of the QC structure is shown in Figure 17. When discharging operations, the main trolley in the QC will first take out the container from the vessels and place containers on the transfer platform (the space area below the contact beam), and then the portal trolley of the QC will transport a container from the transfer platform to the AGV. The loading process is the opposite.

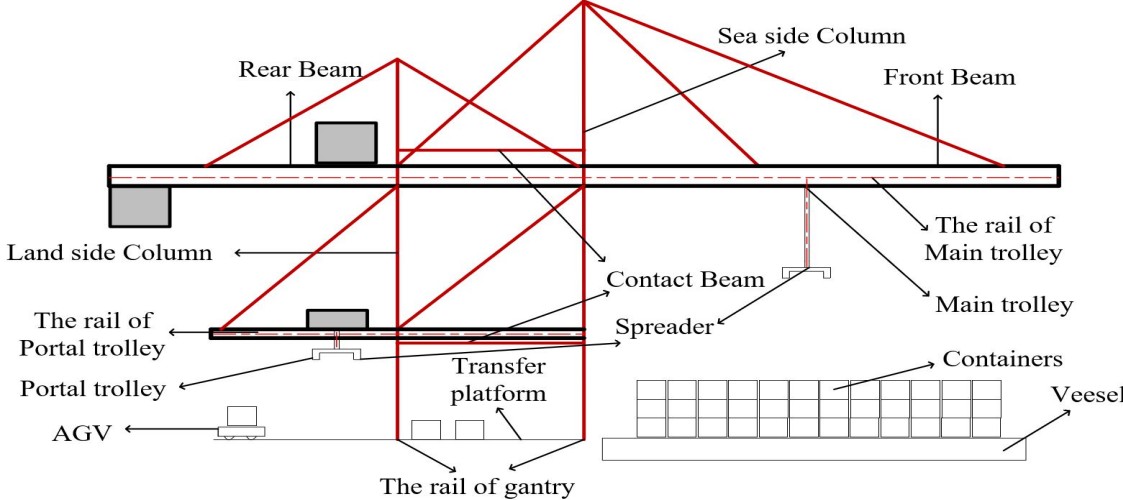

**Figure 17.** The schematic diagram of the QCs.

When modeling QCs, the operation flow of an QC can be expressed by the movement of the main trolley, portal trolley and gantry. In terms of the portal trolley, its operation process can be divided into four categories, namely, the traverse travel of the trolley with/without load and the hoist movement of the trolley with/without load. For the main trolley, except for above four process, the operation of oblique motion (horizontal and vertical movements at the same time) should be included, which is responsible for grabbing the container quickly. Thus, the rail of main trolley is divided into two parts, one for horizontal transportation called out reach and the other for oblique motion called back reach. However, to facilitate the calculation, this paper decomposes the oblique motion into two motions, horizontal and vertical. For the gantry, it only does horizontal movement on the rail when container tasks at assigned bay are completed and moves to another one. Thus, the model of QCs can be seen below.

$$
\begin{aligned}
E_{QC} = {} & \frac{d_m^{tu}}{v_m^{tu}} * P_m^{tu} + \frac{d_m^{hu}}{v_m^{hu}} * P_m^{hu} + \frac{d_m^{tl}}{v_m^{tl}} * P_m^{tl} + \frac{d_m^{hl}}{v_m^{hl}} * P_m^{hl} + \frac{d_g}{v_g} * P_g + \\
& \frac{d_p^{tu}}{v_p^{tu}} * P_p^{tu} + \frac{d_p^{hu}}{v_p^{hu}} * P_p^{hu} + \frac{d_p^{tl}}{v_p^{tl}} * P_p^{tl} + \frac{d_p^{hl}}{v_p^{hl}} * P_p^{hl}
\end{aligned}
\tag{21}
$$

where $d_m^{tu}/d_p^{tu}$ is average moving distance of the QC's main/ portal trolley during traverse traveling without load; $v_m^{tu}/v_p^{tu}$ is average speed of the QC's main/portal trolley in the process of the traverse travel without load; $P_m^{tu}/P_p^{tu}$ is quota power of the QC in the process of the main/portal trolley's traverse travel without load; $d_m^{hu}/d_p^{hu}$ is average moving distance of the QC's main/portal trolley during hoist moving without load; $v_m^{hu}/v_p^{hu}$ is average speed of the QC's main/portal trolley in the process of the hoist movement without load; $P_m^{hu}/P_p^{hu}$ is quota power of the QC in the process of the main/portal trolley's hoist movement without load; $d_m^{tl}/d_p^{tl}$ is average moving distance of the QC's main/portal trolley during traverse traveling with load; $v_m^{tl}/v_p^{tl}$ is average speed of the QC's main/portal trolley in the process of the traverse travel with load; $P_m^{tl}/P_p^{tl}$ is quota power of the QC in the process of the main/portal trolley's traverse travel with load; $d_m^{hl}/d_p^{hl}$ is average moving distance of the QC's main/portal trolley during hoist moving with load; $v_m^{hl}/v_p^{hl}$ is average speed of the QC's main/portal trolley in the process of the hoist movement with load; and $P_m^{hl}/P_p^{hl}$ is quota power of the QC in the process of the main/portal trolley's hoist movement with load.

## 4.5. Energy Consumption Formulation of Vessels

Generally, for the operation of the vessels, it can be divided into three sub-processes: sailing, manoeuvring and berth. During each, carbon emissions may be released by each vessel, from three combustion sources: the main engine, the auxiliary engine and the ship's boiler [47]. In this paper, we only discuss the carbon emissions at the terminal, thus we only consider this process of berth. While most vessels turn off their main engines when in port, their "hoteling" activities require energy from the auxiliary diesel generators. Ship boilers are also working to keep fuel and main engine cylinders warm and avoid damage from low temperature contractions [53]. Nowadays, considering the high pollution of vessels, lots of efforts have been made to realize the climate and environmental goal of green ports, one of which is the on-shore power supply (OPS). OPS, also known as "cold ironing" or shore-side electricity technology, is one of the carbon mitigation strategies by replacing the auxiliary diesel engines with electricity power supplied from shore [54]. For ships connecting to OPS, the air quality and noise will be improved and reduced [55]. Many ports have adopted OPS to reduce emissions in ports, such as Port of Los Angeles and Long Beach, Port of Göteborg and Port of Shanghai.

During berth activity phase, only the auxiliary engines (or OPS) and the ship boilers are in operation. When modeling vessels during berth, the process is divided into two stages: connecting shore power and shore power supply. The auxiliary engines are only involved in the process of shore power connection while the OPS participate in the second stage of work, and the boiler continues to work during this phase.

$$E_v = (l_a * P_a * (t_b - t_c))/\alpha \tag{22}$$

$$D_v = R_d * (P_a * t_c + P_b * t_b)/\rho_d \tag{23}$$

where $E_v$ is power consumption in kWh of the vessel; $D_v$ is diesel consumption in liters of the vessel; $P_b$ is the power of boil, kW; $l_a$ is the load coefficients of auxiliary engines; $p_a$ is the power of auxiliary engines, kW; $t_b$ is the berthing time of vessel, $h$; $t_c$ is the shore power connection time of vessel, $h$; and $\alpha$ is loss factor in transmission from OPS to vessel, $\alpha = 0.92$ [56].

## 4.6. Total Carbon Emission Formulation of the ACT

Through the above analysis, the specific participation of the devices in ACTs can be concluded, as shown in Table 3.

**Table 3.** Devices participation task table.

| Equipment | Operation Task | | | |
|:---:|:---:|:---:|:---:|:---:|
|  | Delivering | Loading | Discharging | Picking-Up |
| OT | 1 | 0 | 0 | 1 |
| ARMG | 1 | 1 | 1 | 1 |
| AGV | 0 | 1 | 1 | 0 |
| QC | 0 | 1 | 1 | 0 |
| Vessel | 0 | 1 | 1 | 0 |

Generally, the containers are divided into three types, namely the import container, the export container and the transfer container (the transfer container can be divided into the transfer container brought by the vessel and the transfer container taken by the vessel); each container will participate in one or more different operational tasks; and the task participation table of containers is shown in Table 4.

**Table 4.** The task participation table of containers.

| Type of Container | Operation Task | | | |
|---|---|---|---|---|
| | Delivering | Loading | Discharging | Picking-Up |
| Export container | 1 | 1 | 0 | 0 |
| Import container | 0 | 0 | 1 | 1 |
| Transfer container (take in) | 0 | 0 | 1 | 0 |
| Transfer container (take away) | 0 | 1 | 0 | 0 |

At container terminals, it is easy to obtain the number of arriving vessels and the number of containers for each type of vessels loading and unloading. Based on these data, the number of containers involved in different operations can be obtained. For example, assuming that X vessels arrive at the terminal within one year, each vessel carries imported containers $x_i$, transfer containers $x_t$, and take out export container $x_e$ and transfer containers $t_t^*$, the annual throughput of the container terminal $Q_h$ and the number of containers involved in operations of delivering, loading, discharging and picking-up $Q_{de}$, $Q_l$, $Q_{di}$, $Q_p$ are as follows. (When calculating terminal throughput, 40 ft and 45 ft containers should be counted as two 2 TEUs.)

$$Q_h = \sum_{x=1}^{X} (x_e + x_i + x_t + x_t^*) \tag{24}$$

$$Q_{de} = \sum_{x=1}^{X} x_e \tag{25}$$

$$Q_l = \sum_{x=1}^{X} (x_e + x_t^*) \tag{26}$$

$$Q_{di} = \sum_{x=1}^{X} (x_i + x_t) \tag{27}$$

$$Q_p = \sum_{x=1}^{X} x_i \tag{28}$$

The total carbon emissions of a container terminal operating equipment $E_T$ can be shown as follows.

$$E_T = \sum_{i=1}^{n} \sum_{j=1}^{m} (\overline{E_{ij}} * f_E + \overline{D_{ij}} * f_D) * Q_j \tag{29}$$

where $\overline{E_{ij}}$ is the average power consumption in kWh of the ith equipment when performing the jth task; $\overline{D_{ij}}$ is the average consumption of diesel in liters of the ith equipment when performing the jth task; $Q_j$ is the number of containers involved in task j; $f_E$ is the emission factor in kilograms of $CO_2$-emission per kWh; and $D_E$ is the emission factor in kilograms of $CO_2$-emission per lit diesel.

Combined with:

$$\overline{E_{ij}} = \overline{E_{ij,V}} + \overline{E_{ij,W}} = \sum_{r=1}^{R} (\frac{l_{ijr}}{v_{ijr}} + \overline{t_{ijr}}) * P_{ijr} \tag{30}$$

where $\overline{E_{ij,V}}$ is the average power consumption in kWh during the ith equipment is moved when performing the jth task; $\overline{E_{ij,W}}$ is the average power consumption in kWh during the ith equipment is waiting when performing the jth task; $l_{ijr}$ is the average moving distance of the ith equipment when performing the jth task in the rth process; $v_{ijr}$ is the average moving speed of the ith equipment when performing the jth task in the rth process; $\overline{t_{ijr}}$ is the average waiting time of the ith equipment when performing the jth task; and $P_{ijr}$ is the quota power of the ith equipment when performing the jth task.

In addition, during the operation, the terminal will have some other energy consumption, resulting in fixed emissions, such as operating platform, employee office area, equipment maintenance area, etc. As the influence on the layout is small and does not change greatly with the layout change, no other considerations are added to the model in this paper.

## 5. Case Study

In this section, a case study is presented to demonstrate the proposed methodology.

### 5.1. Calculate the Average Energy Consumption of Each Equipment

This paper takes an ACT in East China as an example, the shoreline of which is 2350 m. Except for the auxiliary facilities area, about 2100 m is used for storage areas, and the average width of that ACT is 600 m (In fact, there are more than one type of QCs and ARMGs in that ACT, but for the convenience of calculation, this paper selects the type of equipment with the largest number.).

#### 5.1.1. Energy Consumption Formulation of OTs in a Typical ACT

In a typical ACT, the average horizontal travel distance of OTs with load and without load can be regarded as half the length of the yard, and the average vertical travel distance can be regarded as the width of OTs driving area plus the width of the land side exchange area. The waiting time is related to the degree of congestion on the lanes, the speed of the ARMG operation, and other factors, which is relatively complicated. While there is no doubt that the number of lanes is negatively correlated with the waiting time, to simplify the calculation, we assume that there is a perfect number of lanes $n_{Od}^*$, which can make the OTs not to wait. Once the number of lanes is reduced to $n_{Od}$, the waiting time is increased to $(n_{Od}^* - n_{Od})^{(1/3)} * t_{Oi}^*$.

It is worth noting that, in the ACTs, the OT is an important carbon-emitting device, but compared with QC, AGV, etc., the type of OT is greatly different due to the shipping company or other factors. As the type data of OT is difficult to count, and the average carbon emission calculation of OT is complicated in the actual operation, to make the calculations simple, we assume that all the OTs are of Type D. The performance parameters of Type D OTs are shown in Table 5.

**Table 5.** The performance parameters of Type D OTs.

| The speed with load | 20 km/h | The power with load | 200 kw |
|---|---|---|---|
| The speed without load | 35 km/h | The power without load | 160 kw |
| $n_{Od}^*$   20 | $t_{Oi}^*$   1 min | The power during idling | 360 kw |

According to Equation (11), the average power consumption of an OT for transporting a container can be calculated.

#### 5.1.2. Energy Consumption Formulation of ARMGs in a Typical ACT

The sample ACT adopts Type B ARMG with a span of 31 m and lifting height of 19.5 m. The performance parameters of Type B ARMGs are shown in Table 6. Containers differ not only in size but also in weight, and the weight varies greatly according to the difference of goods in the container. Therefore, in this paper, the weight of the container is set to 60 t.

**Table 6.** The performance parameters of Type B ARMGs.

| Rated Load | Under Spreader | | 60 t | |
|---|---|---|---|---|
| Speeds | Main Hoist | hoisting | 60 t load | 24 m/min |
| | | | Empty Spreader | 52 m/min |
| | | lowering | 60 t load | 24 m/min |
| | | | Empty Spreader | 52 m/min |
| | Gantry | Wind speed (lower than) | 16 m/s | 210 m/min |
| | | Wind speed (lower than) | 25 m/s | 100 m/min |
| | Trolley of traverse travel | | | 70 m/min |
| Motors | Main Hoist | | 2 ∗ 400 kw | |
| | Trolley of traverse travel | | 200 kw | |
| | Gantry | | 20 ∗ 14 kw | |

It is assumed that the wind speed is greater than 16 m/s during 30% of the operations, and the average speed of gantry is 177 m/min.

In general, RMG has three configurations: (a) twin cranes or non-crossing cranes, meaning that two RMGs of the same size operate on the same rail and cannot pass each other; (b) dual or crossover cranes, referring to two different sizes RMG (outer and inner cranes) that run on different tracks and can cross each other; or (c) triple cranes, including two twin cranes and one larger crane that move on different tracks and can pass twin cranes [57]. In this paper, we only analyze the first configuration, which means two ARMGs are arranged in one block: one is responsible for the sea side and the other is responsible for the land side. When busy, they can help each other.

ARMGs participate in all operations, including delivering, loading, discharging and picking-up, while in each operation the average running distance of the gantry is different because of different workflows (the detailed description is shown in Appendix B). We assume ARMGs move at normal speed without any interference. Therefore, for each container, the average moving distance of the ARMG in each operation can be represented as $d_{g1}^m = d_{g3}^m = \frac{3}{2}L_b$, $d_{g2}^m = d_{g4}^m = \frac{3}{2}L_b$.

The above parameters are used in Equations (12)–(18) to calculate the average power consumption of ARMGs for transporting a container.

### 5.1.3. Energy Consumption Formulation of AGVs in a Typical ACT

The sample ACT is Type C. The rated load is set as 60 t. The average speed of Type C AGV in the process of traveling with and without load is set as 3.5 m/s and 5.8 m/s, respectively. The quota power of Type C AGV in the process of traveling with/without load is set as 200 KW/160 KW. According to actual observations, AGV generally circulates with three blocks. The $L_t$ is set as 99 m, and the cycle length of the AGV path can be calculated. In fact, the distance traveled by an AGV in the process of traveling with load and the distance traveled without load can be considered equal, which means $l_{Al} = l_{Au} = \frac{1}{2}C$. Then, the average power consumption of an AGV for transporting a container can be calculated.

### 5.1.4. Energy Consumption Formulation of QCs in a Typical ACT

In the sample ACT, the specific process of QCs is shown in Appendix C, and the performance parameters of Type A QCs are shown in Tables 7 and 8.

**Table 7.** The Size parameter of Type A QCs.

| length of rear beam | 30 m | length of front beam | 65 m |
|---|---|---|---|
| Rail Gage | 30 m | Height of the portal rail | 16 m |
| Out reach | 62 m | length of the portal rail | 30 m |
| Back reach | 20 m | Height of lift (above rail) | 41 m |

**Table 8.** The performance parameters of Type A QCs.

| Rated Load | Under Spreader | | | 60 t | |
|---|---|---|---|---|---|
| Speeds | Main Hoist | hoisting | | 60 t load | 75 m/min |
| | | | | Empty Spreader | 150 m/min |
| | | lowering | | 60 t load | 75 m/min |
| | | | | Empty Spreader | 150 m/min |
| | Portal Hoist | hoisting | | 60 t load | 60 m/min |
| | | | | Empty Spreader | 120 m/min |
| | | lowering | | 60 t load | 60 m/min |
| | | | | Empty Spreader | 120 m/min |
| | Main Trolley | | | | 210 m/min |
| | Portal Trolley | | | | 200 m/min |
| | Gantry | | | | 46 m/min |
| Motors | Main Hoist | | | 2 * 560 kw | |
| | Main Trolley | | | 224 kw | |
| | Portal Hoist | | | 2480 kw | |
| | Portal Trolley | | | 200 kw | |
| | Gantry | | | 20 * 17.5 kw | |

According to actual observations, ships have an average of 15 layers: 5 layers on the ship and 10 floors under the ship. The height of a container is 2.4 m and the width of the vessel is 41 m. Assuming a QC processes an average of 300 containers, the average moving distance of QC is 60 m. The above parameters are used in Equation (21) to calculate $\overline{E_{QC}} = 33$ kwh.

5.1.5. Energy Consumption Formulation of Vessels in a Typical ACT

Similar to OTs, the types of ships arriving in ports are diverse and the facilities can vary significantly among different ships. Considering that the vessel is the main source of wharf emissions, this paper takes the ship into account in the calculation. However, it is worth noting that this paper studies the layout of the land area of the wharf, not including the operations planning problems. The carbon emissions of ships at the docks are related to the type of auxiliary facilities and the time of berthing. The berthing time is related to the berth allocation, the quay crane assignment, the quay crane scheduling and other operational plans, but has no influence on the research object of this paper. Therefore, to facilitate the comparison of the equipment on the ACT for better layout of the terminal, this paper simplifies the ship calculation by assuming that the ship type is new Panamax type, thus $P_a = 16,500$ kw, $P_b = 565$ kw [53], and uses the empirical data of previous studies. ($l_a$ can vary significantly among different ships and ports; according to Khersonsky et al. [58], $l_a = 0.63$.)

Before the vessel arrives at the port, the terminal obtains the information of the container vessel from the shipping company or the ship's agent, including the ship's stowage map, manifest, etc., to obtain the loading and unloading task of the container ship, the estimated arrival time, the draft information, etc. Then, in combination with the berths of the terminals and the work of the QCs, berths, QCs and storage areas are arranged for the inbound vessels. After knowing the total number of

containers handled by the vessel, the average emissions of a container from the vessel can be calculated by dividing the total emissions of the ship by the number of containers.

According to the actual data of the terminal, the general connection time is 10 min, the average berth time is 10 h, and the average number of loading and unloading containers of a ship is 1800 TEU. Among them, 20 ft containers and 40 ft containers account for 50% each, ignoring 45 ft containers, which means the number of containers is 1200. There are four QCs serving one ship at the same time on average. According to Equation (22) and (23), the following can be calculated.

$$\overline{E_v} = E_v / n_c = 92.558 \text{ kwh} \tag{31}$$

$$\overline{D_v} = D_v / n_c = 1.667 \text{ L} \tag{32}$$

*5.2. The Calculation of Different Layout Result*

As the average length of OTs/AGV in practice is about 15 m, the turning radius of the OTs is set as 10 m, the distance of OT direct reversing is set as 15.5 m, the length of OT slots in the loading and unloading area is set as 15.5 m, and the safe distance of the land side exchange area is set as 0.4 m. According to Equation (2), $W_{LE} = 41.4$ m can be calculated. Similarly, the turning radius of the AGVs is set as 10 m, the distance of AGV direct reversing is set as 15.5 m, the length of AGV mates is set as 15.5 m, and the safe distance of the sea side exchange area is set as 0.4 m. Thus, $W_{SE} = 41.4$ m, $W_b = 35.5$ m can be calculated according to Equation (3). According to the performance data of QC in Table 7, $W_{QC} = 60.5$ m can be calculated. In that ACT, the AGV driving lane is set as 4 m, $W_{Ad} = n_{Ad} * 4$, the OT driving lane is set as 3.5 m, $W_{Od} = (n_{Od} - 1) * 3.5 + w_{Odd}$, as the length of 20-foot container is generally equal to 6.1 m, thus $W_S = n_b * l_c + (n_b - 1) * 0.3 + 0.8$ can be calculated according to Equations (7) and (8).

According to the above description and Equation (1), we can get the following formula.

$$(n_{Od} - 1) * 3.5 + w_{Odd} + n_{Ad} * 4 + n_b * 6.4 = 420.7 \tag{33}$$

where $n_{Ad}$, $n_b$, and $n_{Od}$ are used as variables, while $w_{Odd}$ is the dependent variable.

Based on the actual operational data of the terminal, the annual throughput of the terminal is set as 600 million TEU, of which loading, discharging, delivering and picking-up account for 3 million TEU, 3 million TEU, 1.5 million TEU and 1.5 million TEU, respectively, and the transfer containers for 1.5 million TEU. When it comes to container types, 20 ft and 40 ft account for 50% each, and the 45 ft container is ignored. Thus, $Q_{de} = 100$ million, $Q_l = 200$ million, $Q_{di} = 200$ million, and $Q_p = 100$ million can be calculated.

The imbalance coefficient of container vehicles arriving at the port $K_{BV}$ is set as 2.5, the number of working days (d) of the container terminal $T_{yk}$ is set as 360 d, the working time of the gate $T_d$ is set as 16 h, the number of vehicles passing through a single lane $p_d$ is set as 45 vehicles/h, and the average vehicle capacity $q_c$ is set as 1.5 TEU/car. The sample ACT only involves the water and water transfer, land and water transfer, the percentage of the total container volume of dismantling and water transfer containers within the inland area of the container terminal gate accounts for the annual volume of the terminal $K_b$ is set as 50%. Thus, $N_d \approx 19$ can be calculated according to Equation (9).

In this ACT, only the gates of the inbound gate and the exit gate are considered, assuming that the average diversions of the inbound and exit gates, which means the number of OTs driving lanes of the inbound and exit gates, are equal to each other, thus should be satisfied $n_{Od} \gg \frac{1}{2} N_s = 9$ in design. However, too many driving lanes may appear unreasonable, thus this article limits the number of OTs driving lanes in the terminal to the total number of driving lanes through gates.

Similarly, if $K'_{BV} = 1.2$, $T'_d = 20$, $N'_d \approx 14$ can be calculated, $\frac{1}{2} N'_d = 7 \ll n_{Ad} \ll 14$ should be satisfied.

As rail gauge of the ARMG is 31 m and 4 m between adjacent blocks, a total of 60 blocks are set in the ACT and 10 rows are set in a block, and the number of ground slots in the container terminal yard is $N_s = 600 * n_b$.

According to the actual operational data of the terminal, the container imbalance factor of the yard is set as 1.25, the yard capacity utilization rate is set as $A_S = 70\%$, and the average stockpile period is set as $t_{dc} = 6$ days, thus, according to Equations (17) and (15), the following can be calculated.

$$\overline{h} = N_1 * A_S = (Q_h t_{dc} K_{BK})/(T_{yk} N_s) = 208.33/n_b. \tag{34}$$

$$N_r = 41/80\overline{h} - 19/80 = 106.77/n_b - 19/80 \tag{35}$$

As the container storage height $N_1$ does not exceed six in practice, $n_b \gg 50$ can be calculated according to above formula.

After the above discussion, the constraint conditions are as follows:

- $7 \leq n_{Ad} \leq 14$
- $9 \leq n_{Od} \leq 18$
- $3.5 \leq w_{Odd} \leq 7$
- $n_b \geq 50$
- $7(n_{Od} - 1) * 3.5 + w_{Odd} + n_{Ad} * 4 + n_b * 6.4 = 420.7$

The layout combination that satisfies the above constraints can be calculated (Table A1).

Based on the above carbon emission calculation model, the average carbon emissions of each type of equipment when processing a single container can also be calculated. According to PENG [59], the $CO_2$ emission factors for diesel is 2.65 kg/L. The sample port is located in East China, thus the power $CO_2$ emission factor of this regional grid is 0.8367 kg/kwh. Thus, the total carbon emissions can be calculated as shown in Tables A2 and A3.

As can be seen in Table A3, the minimum $CO_2$ emission is $1163.222 \times 10^6$ kg (the first combination), the most is $1186.454 \times 10^6$ kg (the 17th combination). Compared to the first case, the 17th case will increase $CO_2$ emissions by 2%. Except for the emissions of QCs and vessels, which have no effect on the layout of ACTS analyzed in this paper, this percentage reaches 4%. This means that, if the port is properly laid out, it can reduce carbon emissions by $23.232 \times 10^6$ kg per year. From the perspective of the entire life cycle of ACT, the value would be even larger.

### 5.3. Result Analysis

As discussed above, there is a balance of relationships within each region of ACTs, and there are constraints and balances between regions. Below, we carefully analyze from partial to global.

### 5.3.1. Analysis of the Od Area

The main equipment in the Od is the OTs, and the OTs mainly consumes diesel to emit carbon dioxide. The variable associated with the Od area is the number of OTs driving lanes $n_{Od}$. According to the operational data, the relationship between the relevant diesel consumption and $n_{Od}$ is shown in Figure 18.

It can be found that more OTs driving lanes leads to a wider Od area, and slightly increased diesel consumption caused by the vehicle travel, while the diesel consumption caused by the idle travel will be significantly reduced, resulting in a reduction in the total amount of average carbon emissions for transporting a container. Considering that, with the development of ACT, the throughput will increase, and the number of OTs entering and leaving the yard will increase, when designing ACT, there should be as many OTs driving lanes as possible.

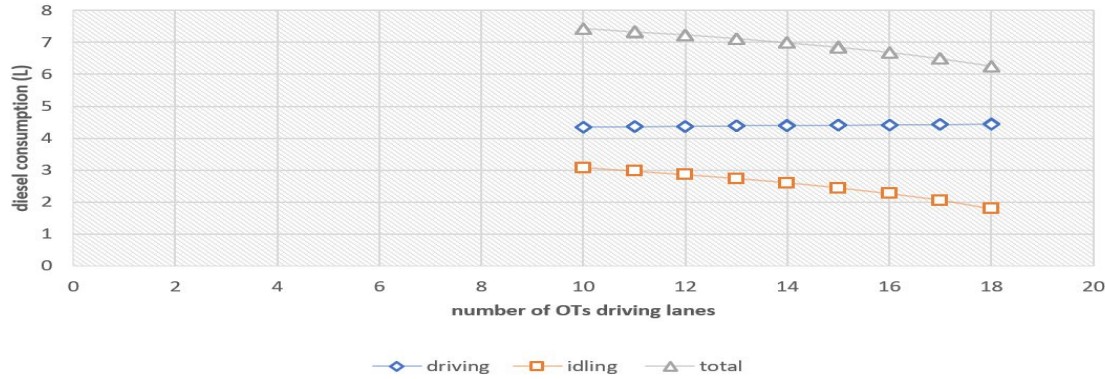

**Figure 18.** Average diesel consumption of an OT in ACT under different lane numbers.

### 5.3.2. Analysis of the Ad Area

The main equipment in the Ad is AGVs, which generate carbon dioxide mainly by power consumption. The variable associated with the Ad area is the number of AGV driving lanes $n_{Ad}$. According to the operational data, the relationship between the relevant power consumption and $n_{Od}$ is shown in Figure 19.

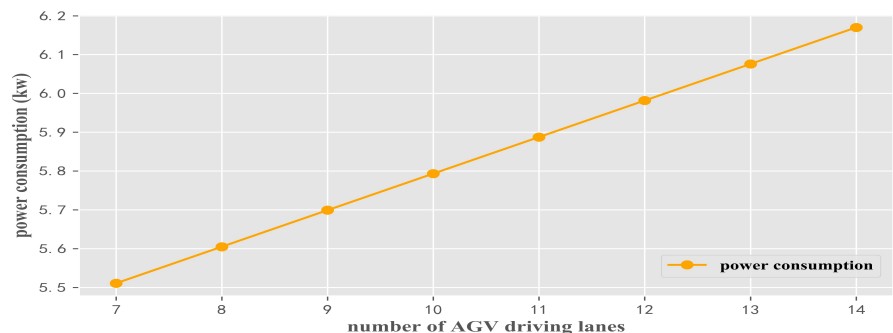

**Figure 19.** Average power consumption of an AGV during one operation under different $n_{Ad}$.

As the power consumption of the AGVs is extremely small, only the power consumption when the vehicle is running is considered. Therefore, it can be seen that more lanes lead to longer average travel distance of the AGV, resulting in more power consumption. Considering the low emission of AGV compared with other equipment and the great advantage of having buffer zones as a temporary parking point, when designing ACT, the number of lanes of AGV can be reduced as much as possible on the basis of satisfying the traffic flow of AGV.

### 5.3.3. Analysis of the Storage Yard Area

The relationship in the storage yard area is relatively complicated. In the case where the throughput is certain and the stack height is limited, there is no doubt that, as the number of bays increases, the average number of re-handles will gradually decrease, as shown in Figure 20.

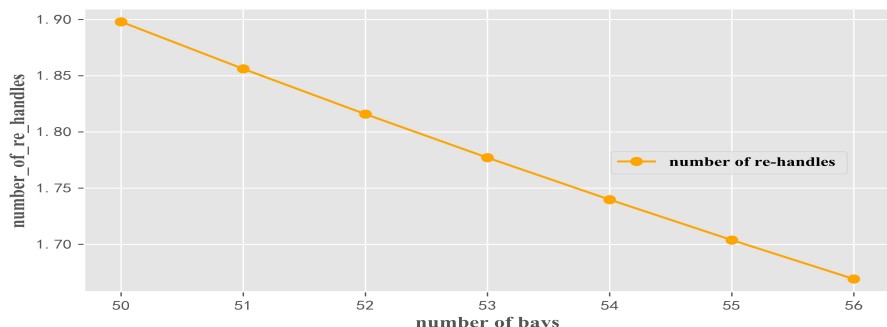

**Figure 20.** The average number of re-handles under different bays.

For the main equipment ARMG in the yard, as the number of bays $n_b$ increases, the average moving distance of the gantry increases. Taking delivering and loading operation as an example, the average driving distance of the gantry varies with $n_b$, as shown in Figure 21a.

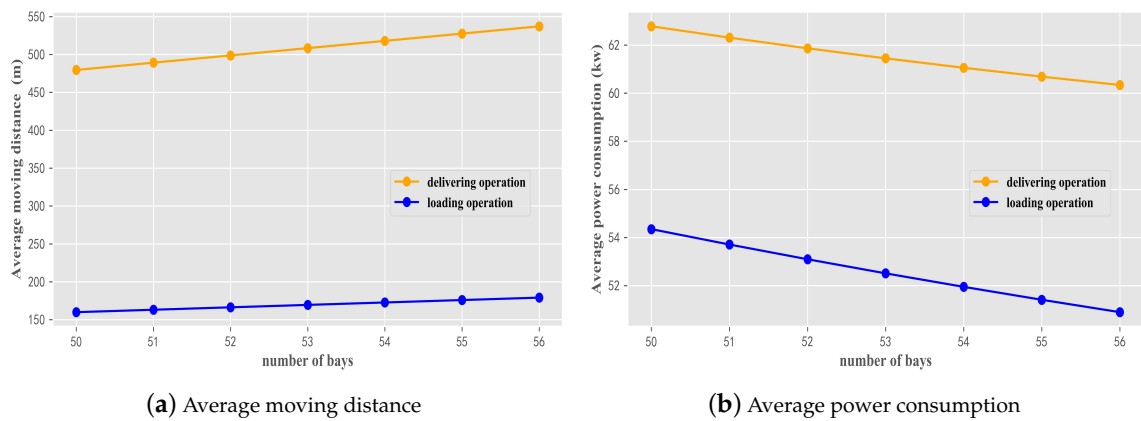

(**a**) Average moving distance　　　　　　　　　　　(**b**) Average power consumption

**Figure 21.** Average power consumption/Average moving distance of ARMG gantry during one operation under different bays.

However, the number of re-handles being reduced causes the average moving distance of the trolley to be reduced, thus the movement of gantry and trolley will be restricted and balanced due to changes in the number of bays. For the number of bays meeting the above conditions, the resulting average power consumption of ARMG during one operation is shown in Figure 21b.

It can be seen that, within the specified area of the number of bays, as the $n_b$ increases, the power consumption of ARMG will gradually decrease. It is worth noting that, in the subsequent development of the wharf, throughput will gradually increase, while the storage height is still limited to today's trend, and more number of bays means more ground slots and greater throughput. Therefore, it is necessary to design the length of the block long enough when designing the ACT in terms of carbon emissions and throughput.

### 5.3.4. Analysis of Total Carbon Emissions Based on Correlation Analysis

Based on the above table, it is difficult to find the effect of each variable and total carbon emissions. Correlation analysis was conducted to explore the law.

In this paper, Canonical Correlation Analysis (CCA) is used, which was proposed by Hotelling in 1936 and matured in the 1970s. In fact, CCA studies the correlation between two sets of variables $X = (X_1, X_2, X_3, ..., X_n)$ and $Y = (Y_1, Y_2, Y_3, ..., Y_m)$. It is usually measured by the correlation coefficient, as in the following formula,

$$\rho_{xy} = \frac{Cov(X, Y)}{\sqrt{Var(X)}\sqrt{Var(Y)}} \tag{36}$$

By calculation, $\rho_{(n_{Od}E_T)} = 0.100$, $\rho_{(n_{Ad}E_T)} = 0.887$, $\rho_{(n_bE_T)} = -0.843$, and the scatter plot is shown in Figure 22.

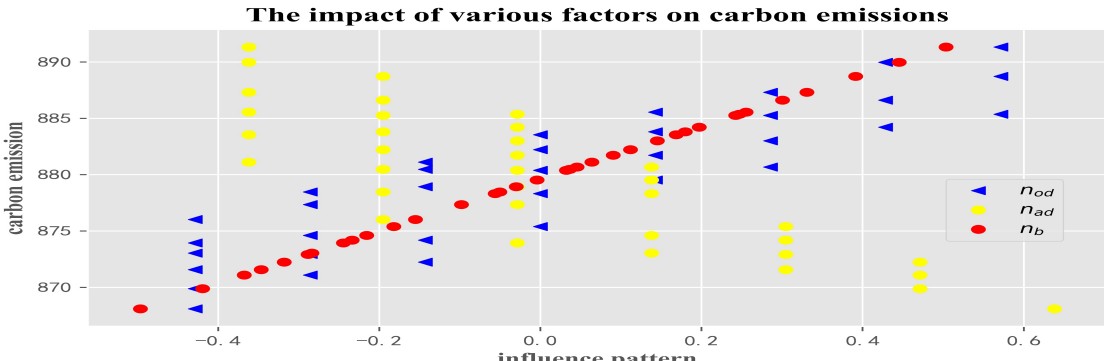

**Figure 22.** Correlation analysis of three decision variables.

From the results, we can see that $n_{Ad}$ and $n_b$ have a greater correlation with carbon emissions, among which $n_{Ad}$ is the largest, because the number of lanes of AGV directly affects the AGV path, thus directly affects carbon emissions, while there are constraints and balances in Od and storage yard.

The proportion of average single and total $CO_2$ generated by each device is shown in Figure 23.

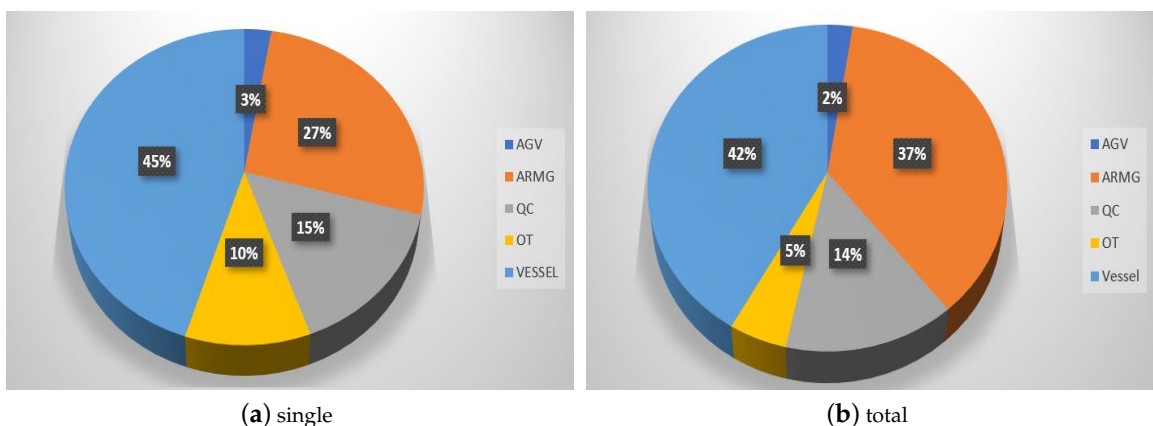

(**a**) single        (**b**) total

**Figure 23.** Proportion of carbon emissions from various equipment in ACT.

As shown in Figure 23, the order of carbon emission can be expressed as follows, whether it is the average carbon emission of dealing with a single container or the total carbon emission of handling containers in one year.

$$Vessel > ARMG > QC > OT > AGV \tag{37}$$

Among them, ARMG, vessel and QC account for a larger proportion, while OT and AGV account for a rather small proportion. However, compared with the average emissions of handling single container, the proportion of each equipment in the total emissions has changed greatly, which is because the number of containers participated in by different equipment is different in the terminal. In addition, it can be found that, in the ACT, the main source of carbon emissions is vessel, as ships have high power requirements. ARMG is the second largest source of carbon emissions, which participates in the processing of the largest number of containers. Therefore, the layout design should focus on the area served by ARMGs, that is, the yard area.

In short, from the point of view of the sustainable development of the terminal, in the design of a typical ACT, the yard area should be considered first. To minimize carbon emissions and take into account the sustained growth of throughput, the block area should be long enough. In addition,

the number of OTs driving lanes should be as large as possible, and the number of lanes in AGV driving area should be as small as possible while meeting its traffic flow. Other areas can be considered constant at design time depending on the selection of the actual equipment.

## 6. Conclusions and Discussion

Through the aforementioned analysis, the main contributions of this paper can be summarized as follows:

1.  A novel total $CO_2$ emission model of equipment in ACT based on equipment periodic operation process is presented, which can help dock designers calculate the carbon emissions of a wharf according to throughput and equipment type when designing terminal. The calculation results can help a terminal company make sound decisions before operating.

    Based on the analysis of data of the case study, the following suggestions were obtained:

    (1) Vessels are the largest carbon emission equipment in a wharf, and the direction of reducing carbon emissions from ports in the future can focus on improving the emissions of ships, such as using automatic mooring system (AMS), replacing oil-driven ships with electric ones or reducing time of berth by improving terminal efficiency.

    (2) The reason ARMG is the main source of $CO_2$ emission is because, in this typical ACT, the horizontal transport equipment does not enter the container area, which leads to a larger periodic operation path of ARMG's gantry, and the number of containers that ARMG participates in processing is the largest. Future research on reducing $CO_2$ emissions from wharfs can focus on improving the operation process of ARMG to reduce its periodic operation path.

2.  A conceptual model considering the interaction between different areas of the typical ACTs when the width and length of the terminal are fixed is established. The width of an ACT can be regarded as the circumference of a circle, and the width of some areas in ACT can be regarded as a fixed value, while the width of Od, Ad and storage yard area is not fixed and can be freely scaled on the circle. The result of left and right scaling will lead to the change of the periodic operation flow of both the equipment in the area and the other areas.

    After analysis, it was found that:

    (1) In the Od area, the more driving lanes there are, the more carbon emissions are generated by OTs driving, while the carbon emissions generated by idling are reduced, and finally the total carbon emissions are reduced.

    (2) In the Ad region, as the number of driving lanes increases, the more carbon emissions are generated by AGV driving, as this paper directly ignores the little power consumption during waiting, and finally increases the power consumption, so that the total carbon emissions increase slightly.

    (3) In the SY area, as the number of bays increases, average number of re-handles decreases, the average driving path of the trolley shortens, and the average driving distance of the gantry lengthens. However, with the increase of number of bays, the total power consumption and total carbon emissions shows a downward trend.

    (4) In the typical ACTs, the number of OTs driving lanes $n_{Od}$, the number of AGVs driving lanes $n_{Ad}$ and bays in the storage yard $n_b$ will affect the average operating distance of equipment in the region, which will affect the total carbon emissions, while $n_{Ad}$ has the greatest correlation with the total carbon emissions, and $n_b$ has a negative correlation with the total carbon emissions.

    (5) The typical ACT layout design sequence considering the sustainable development of a wharf is as follows: the block area is designed long enough; the OT driving area is appropriately increased; and the AGV driving area is properly reduced.

In this paper, we mainly discuss the mutual influence of regional distance variation and the impact on total carbon emissions when the width of dock is fixed in typical ACT, which can be extended to more directions in the future. Some examples are listed below:

- When the length of the yard is fixed, the impact of changes in the layout of each area (such as the OT/AGVs entry and non-entry yard) on the total carbon emissions can be discussed, or the layout design under atypical shape can be discussed.
- There are three variables ($n_{Ad}$, $n_{Od}$, and $n_B$) in this paper, but throughput, transfer rate, equipment type, total size of terminal, etc. could also be considered. These parameters will affect the final total carbon emissions, too. Among them, throughput and transfer rate affect the number of containers handled by equipment, and equipment type and total size of terminal affect the average operation path of the equipment.
- This paper mainly considers the impact of different layouts on the total carbon emissions, but different layouts will also affect the efficiency of equipment operation and the quality of service of the terminal, which could be considered in depth in the future.
- The model constructed in this paper is a single objective function, and, in the future, a multi-objective function covering various factors such as cost and benefit could be constructed. This article mainly discusses the size layout in the typical ACTs, while "process design + area size", "equipment type + site size", etc. could be considered in further studies.

**Author Contributions:** N.W. contributed tothe theoretical approaches, validation, and preparing the article; J.Y. contributed to the formal analysis; D.C. and X.S contributed to the supervision; and Y.G. contributed to the article editing.

**Funding:** This research was funded by Shanghai Science and Technology Committee (grant numbers 16DZ2349900 and 16DZ1201402), and National Natural Science Foundation of China (61540045 and 71602114).

**Acknowledgments:** The authors would like to thank the Shanghai Science and Technology Committee and National Natural Science Foundation of China for providing the facilities and financial support.

**Conflicts of Interest:** The authors declare no conflict of interest.

## Abbreviations

The following abbreviations are mainly used in Section 3:

| | | | |
|---|---|---|---|
| QC | quay crane | AGV | Automatic Guided Vehicle |
| OT | outside truck | ARMG | Automatic Rail-Mounted Gantry Crane |
| Od | the OTs driving lane | $W_{Od}$ | the width of the OTs driving lane |
| Ad | the AGVs driving lane | $W_{Ad}$ | the width of the AGVs driving lane |
| SE | the sea side exchange area | $W_{SE}$ | the width of the sea side exchange area |
| LE | the land side exchange area | $W_{LE}$ | the width of the land side exchange area |
| SY | the storage yard area | $W_s$ | the width of the storage yard |
| $n_{Od}$ | the number of OT driving lanes | $W_b$ | the width of the buffer zone |
| $n_{Ad}$ | the number of AGV driving lanes | $W_{QC}$ | the width of the QCs operation area |
| $n_b$ | the number of bays | $L$ | the total length of the QCs operation area |
| ACT | Automated Container Terminal | $W$ | the total width of the ACT |

The following abbreviations are mainly used in Section 4:

| | |
|---|---|
| $\overline{D_{OT}}$ | the average diesel consumption in liters of an OT transporting a container in the ACT |
| $D_{OT}$ | the diesel consumption in liters of OTs in ACTs |
| $l_{Ol}/l_{Ou}$ | the distance traveled by a loading/unloading OT to deliver the container at the terminal (km) |
| $v_{Ol}/v_{Ou}$ | the average speed when the OT is loaded/unloaded (km/h) |
| $t_{oi}$ | the total waiting time when an OT is idling at the terminal (min) |
| $P_{Ol}/P_{Ou}$ | quota power of the OT in the process of traveling with/without load |
| $P_{Oi}$ | quota power of the OT in the process of idling |
| $R_{OT}$ | the diesel consumption rate of OTs, 0.2 kg/kWh |
| $\rho_d$ | the density of diesel, 0.84 kg/L. |
| $\overline{E_{ARMG/AGV/QC}}$ | the average power consumption of an(a) ARMG/AGV/QC handling a container in ACTs |
| $E_{ARMG/AGV/QC}$ | power consumption in kWh of the ARMG/AGV/QC |
| $E^T_{OT/AGV/QC/ARMG/Vessel}$ | the total $CO_2$ emission of OTs/AGVs/QCs/ARMGs/Vessels handling containers in ACTs |
| $d^m_g$ | average moving distance of the ARMG's gantry in the process of gantry operation |
| $v^m_g$ | average speed of the ARMG's gantry in the process of gantry operation |
| $P^m_g$ | quota power of the ARMG in the process of gantry operation |
| $t^i_g$ | average waiting time of the ARMG in the process of gantry idle |
| $P^i_g$ | quota power of the ARMG in the process of gantry idle |
| $d^{tu}_t/d^{tl}_t$ | average moving distance of the ARMG's trolley during traverse traveling without/with load |
| $v^{tu}_t/v^{tl}_t$ | average speed of the ARMG's trolley in the process of the traverse travel without/with load |
| $P^{tu}_t/P^{tl}_t$ | quota power of the ARMG in the process of the traverse travel without/with load |
| $d^{hu}_t/d^{hl}_t$ | average moving distance of the ARMG'S trolley during hoist moving without/with load |
| $v^{hu}_t/v^{hl}_t$ | average speed of the trolley of ARMG in the process of the hoist movement without/with load |
| $P^{hu}_t/P^{hl}_t$ | quota power of the ARMG in the process of the hoist movement without/with load |
| $d_{Al}/d_{Au}$ | the distance traveled by an AGV in the process of traveling with/without load |
| $v_{Al}/v_{Au}$ | the average speed when the AGV in the process of traveling with/without load |
| $P_{Al}/P_{Au}$ | quota power of the AGV in the process of traveling with/without load |
| $t_{Ai}$ | the total waiting time when an AGV is idling at the terminal(min) |
| $P_{Ai}$ | quota power of the AGV in the process of idling |
| $d_g$ | average moving distance of the gantry of QC |
| $v_g$ | average speed of the gantry of QC |
| $P_g$ | quota power of the QC in the process of the gantry moving |
| $d^{tu}_m/d^{tu}_p$ | average moving distance of the QC's main/ portal trolley during traverse traveling without load |
| $v^{tu}_m/v^{tu}_p$ | average speed of the QC's main/portal trolley in the process of the traverse travel without load |
| $P^{tu}_m/P^{tu}_p$ | quota power of the QC in the process of the main/portal trolley's traverse travel without load |
| $d^{hu}_m/d^{hu}_p$ | average moving distance of the QC's main/portal trolley during hoist moving without load |
| $v^{hu}_m/v^{hu}_p$ | average speed of the QC's main/portal trolley in the process of the hoist movement without load |
| $P^{hu}_m/P^{hu}_p$ | quota power of the QC in the process of the main/portal trolley's hoist movement without load |
| $d^{tl}_m/d^{tl}_p$ | average moving distance of the QC's main/portal trolley during traverse traveling with load |
| $v^{tl}_m/v^{tl}_p$ | average speed of the QC's main/portal trolley in the process of the traverse travel with load |
| $P^{tl}_m/P^{tl}_p$ | quota power of the QC in the process of the main/portal trolley's traverse travel with load |
| $d^{hl}_m/d^{hl}_p$ | average moving distance of the QC's main/portal trolley during hoist moving with load |
| $v^{hl}_m/v^{hl}_p$ | average speed of the QC's main/portal trolley in the process of the hoist movement with load |
| $P^{hl}_m/P^{hl}_p$ | quota power of the QC in the process of the main/portal trolley's hoist movement with load |
| $E_v$ | the power consumption in kWh of vessel handling a container during berth |
| $D_v$ | the diesel consumption in liters of vessel handling a container during berth |

## Appendix A. The Detailed Calculation Results of the Case Terminal

**Table A1.** The appropriate layout combination of sample ACT.

| Number of Cases | $n_{Od}$ | $n_{Ad}$ | $n_b$ | $w_{Odd}$ (m) | $W_{Od}$ (m) | $W_{Ad}$ (m) | $W_s$ (m) | $W_{LE}$ (m) | $W_{SE}$ (m) | $W_b$ (m) | $W_{QC}$ (m) | $W$ (m) |
|---|---|---|---|---|---|---|---|---|---|---|---|---|
| 1 | 9 | 7 | 56 | 6.3 | 34.3 | 28 | 358.9 | 41.4 | 41.4 | 35.5 | 60.5 | 600 |
| 2 | 9 | 9 | 55 | 4.7 | 32.7 | 36 | 352.5 | 41.4 | 41.4 | 35.5 | 60.5 | 600 |
| 3 | 9 | 12 | 53 | 5.5 | 33.5 | 48 | 339.7 | 41.4 | 41.4 | 35.5 | 60.5 | 600 |
| 4 | 9 | 14 | 52 | 3.9 | 31.9 | 56 | 333.3 | 41.4 | 41.4 | 35.5 | 60.5 | 600 |
| 5 | 10 | 8 | 55 | 5.2 | 36.7 | 32 | 352.5 | 41.4 | 41.4 | 35.5 | 60.5 | 600 |
| 6 | 10 | 10 | 54 | 3.6 | 35.1 | 40 | 346.1 | 41.4 | 41.4 | 35.5 | 60.5 | 600 |
| 7 | 10 | 11 | 53 | 6 | 37.5 | 44 | 339.7 | 41.4 | 41.4 | 35.5 | 60.5 | 600 |
| 8 | 10 | 13 | 52 | 4.4 | 35.9 | 52 | 333.3 | 41.4 | 41.4 | 35.5 | 60.5 | 600 |
| 9 | 10 | 14 | 51 | 6.8 | 38.3 | 56 | 326.9 | 41.4 | 41.4 | 35.5 | 60.5 | 600 |
| 10 | 11 | 7 | 55 | 5.7 | 40.7 | 28 | 352.5 | 41.4 | 41.4 | 35.5 | 60.5 | 600 |
| 11 | 11 | 9 | 54 | 4.1 | 39.1 | 36 | 346.1 | 41.4 | 41.4 | 35.5 | 60.5 | 600 |
| 12 | 11 | 10 | 53 | 6.5 | 41.5 | 40 | 339.7 | 41.4 | 41.4 | 35.5 | 60.5 | 600 |
| 13 | 11 | 12 | 52 | 4.9 | 39.9 | 48 | 333.3 | 41.4 | 41.4 | 35.5 | 60.5 | 600 |
| 14 | 12 | 8 | 54 | 4.6 | 43.1 | 32 | 346.1 | 41.4 | 41.4 | 35.5 | 60.5 | 600 |
| 15 | 12 | 11 | 52 | 5.4 | 43.9 | 44 | 333.3 | 41.4 | 41.4 | 35.5 | 60.5 | 600 |
| 16 | 12 | 13 | 51 | 3.8 | 42.3 | 52 | 326.9 | 41.4 | 41.4 | 35.5 | 60.5 | 600 |
| 17 | 12 | 14 | 50 | 6.2 | 44.7 | 56 | 320.5 | 41.4 | 41.4 | 35.5 | 60.5 | 600 |
| 18 | 13 | 7 | 54 | 5.1 | 47.1 | 28 | 346.1 | 41.4 | 41.4 | 35.5 | 60.5 | 600 |
| 19 | 13 | 10 | 52 | 5.9 | 47.9 | 40 | 333.3 | 41.4 | 41.4 | 35.5 | 60.5 | 600 |
| 20 | 13 | 12 | 51 | 4.3 | 46.3 | 48 | 326.9 | 41.4 | 41.4 | 35.5 | 60.5 | 600 |
| 21 | 13 | 13 | 50 | 6.7 | 48.7 | 52 | 320.5 | 41.4 | 41.4 | 35.5 | 60.5 | 600 |
| 22 | 14 | 8 | 53 | 4 | 49.5 | 32 | 339.7 | 41.4 | 41.4 | 35.5 | 60.5 | 600 |
| 23 | 14 | 9 | 52 | 6.4 | 51.9 | 36 | 333.3 | 41.4 | 41.4 | 35.5 | 60.5 | 600 |
| 24 | 14 | 11 | 51 | 4.8 | 50.3 | 44 | 326.9 | 41.4 | 41.4 | 35.5 | 60.5 | 600 |
| 25 | 15 | 7 | 53 | 4.5 | 53.5 | 28 | 339.7 | 41.4 | 41.4 | 35.5 | 60.5 | 600 |
| 26 | 15 | 8 | 52 | 6.9 | 55.9 | 32 | 333.3 | 41.4 | 41.4 | 35.5 | 60.5 | 600 |
| 27 | 15 | 10 | 51 | 5.3 | 54.3 | 40 | 326.9 | 41.4 | 41.4 | 35.5 | 60.5 | 600 |
| 28 | 15 | 12 | 50 | 3.7 | 52.7 | 48 | 320.5 | 41.4 | 41.4 | 35.5 | 60.5 | 600 |
| 29 | 16 | 9 | 51 | 5.8 | 58.3 | 36 | 326.9 | 41.4 | 41.4 | 35.5 | 60.5 | 600 |
| 30 | 16 | 11 | 50 | 4.2 | 56.7 | 44 | 320.5 | 41.4 | 41.4 | 35.5 | 60.5 | 600 |
| 31 | 17 | 7 | 52 | 3.9 | 59.9 | 28 | 333.3 | 41.4 | 41.4 | 35.5 | 60.5 | 600 |
| 32 | 17 | 8 | 51 | 6.3 | 62.3 | 32 | 326.9 | 41.4 | 41.4 | 35.5 | 60.5 | 600 |
| 33 | 17 | 10 | 50 | 4.7 | 60.7 | 40 | 320.5 | 41.4 | 41.4 | 35.5 | 60.5 | 600 |
| 34 | 18 | 7 | 51 | 6.8 | 66.3 | 28 | 326.9 | 41.4 | 41.4 | 35.5 | 60.5 | 600 |
| 35 | 18 | 9 | 50 | 5.2 | 64.7 | 36 | 320.5 | 41.4 | 41.4 | 35.5 | 60.5 | 600 |

**Table A2.** The average energy consumption of each equipment in different situations.

| Number of Cases | $\overline{E_{QC}}$ (kw) | $\overline{E_{AGV}}$ (kw) | $\overline{E_v}$ (kw) | $\overline{E^1_{ARMG}}$ (kw) | $\overline{E^2_{ARMG}}$ (kw) | $\overline{E^3_{ARMG}}$ (kw) | $\overline{E^4_{ARMG}}$ (kw) | $\overline{D_v}$ (L) | $\overline{D_{OT}}$ (L) |
|---|---|---|---|---|---|---|---|---|---|
| 1 | 33.000 | 5.511 | 92.588 | 60.340 | 50.897 | 60.340 | 50.897 | 1.667 | 7.516 |
| 2 | 33.000 | 5.699 | 92.588 | 60.687 | 51.412 | 60.687 | 51.412 | 1.667 | 7.511 |
| 3 | 33.000 | 5.982 | 92.588 | 61.448 | 52.511 | 61.448 | 52.511 | 1.667 | 7.514 |
| 4 | 33.000 | 6.170 | 92.588 | 61.865 | 53.097 | 61.865 | 53.097 | 1.667 | 7.508 |
| 5 | 33.000 | 5.605 | 92.588 | 60.687 | 51.412 | 60.687 | 51.412 | 1.667 | 7.425 |
| 6 | 33.000 | 5.793 | 92.588 | 61.056 | 51.950 | 61.056 | 51.950 | 1.667 | 7.420 |
| 7 | 33.000 | 5.887 | 92.588 | 61.448 | 52.511 | 61.448 | 52.511 | 1.667 | 7.428 |
| 8 | 33.000 | 6.076 | 92.588 | 61.865 | 53.097 | 61.865 | 53.097 | 1.667 | 7.422 |
| 9 | 33.000 | 6.170 | 92.588 | 62.308 | 53.709 | 62.308 | 53.709 | 1.667 | 7.431 |
| 10 | 33.000 | 5.511 | 92.588 | 60.687 | 51.412 | 60.687 | 51.412 | 1.667 | 7.333 |
| 11 | 33.000 | 5.699 | 92.588 | 61.056 | 51.950 | 61.056 | 51.950 | 1.667 | 7.327 |
| 12 | 33.000 | 5.793 | 92.588 | 61.448 | 52.511 | 61.448 | 52.511 | 1.667 | 7.336 |
| 13 | 33.000 | 5.982 | 92.588 | 61.865 | 53.097 | 61.865 | 53.097 | 1.667 | 7.330 |
| 14 | 33.000 | 5.605 | 92.588 | 61.056 | 51.950 | 61.056 | 51.950 | 1.667 | 7.227 |
| 15 | 33.000 | 5.887 | 92.588 | 61.865 | 53.097 | 61.865 | 53.097 | 1.667 | 7.230 |
| 16 | 33.000 | 6.076 | 92.588 | 62.308 | 53.709 | 62.308 | 53.709 | 1.667 | 7.224 |
| 17 | 33.000 | 6.170 | 92.588 | 62.779 | 54.349 | 62.779 | 54.349 | 1.667 | 7.232 |
| 18 | 33.000 | 5.511 | 92.588 | 61.056 | 51.950 | 61.056 | 51.950 | 1.667 | 7.116 |
| 19 | 33.000 | 5.793 | 92.588 | 61.865 | 53.097 | 61.865 | 53.097 | 1.667 | 7.119 |
| 20 | 33.000 | 5.982 | 92.588 | 62.308 | 53.709 | 62.308 | 53.709 | 1.667 | 7.114 |
| 21 | 33.000 | 6.076 | 92.588 | 62.779 | 54.349 | 62.779 | 54.349 | 1.667 | 7.122 |
| 22 | 33.000 | 5.605 | 92.588 | 61.448 | 52.511 | 61.448 | 52.511 | 1.667 | 6.988 |
| 23 | 33.000 | 5.699 | 92.588 | 61.865 | 53.097 | 61.865 | 53.097 | 1.667 | 6.996 |
| 24 | 33.000 | 5.887 | 92.588 | 62.308 | 53.709 | 62.308 | 53.709 | 1.667 | 6.991 |
| 25 | 33.000 | 5.511 | 92.588 | 61.448 | 52.511 | 61.448 | 52.511 | 1.667 | 6.849 |
| 26 | 33.000 | 5.605 | 92.588 | 61.865 | 53.097 | 61.865 | 53.097 | 1.667 | 6.857 |
| 27 | 33.000 | 5.793 | 92.588 | 62.308 | 53.709 | 62.308 | 53.709 | 1.667 | 6.851 |
| 28 | 33.000 | 5.982 | 92.588 | 62.779 | 54.349 | 62.779 | 54.349 | 1.667 | 6.846 |
| 29 | 33.000 | 5.699 | 92.588 | 62.308 | 53.709 | 62.308 | 53.709 | 1.667 | 6.690 |
| 30 | 33.000 | 5.887 | 92.588 | 62.779 | 54.349 | 62.779 | 54.349 | 1.667 | 6.685 |
| 31 | 33.000 | 5.511 | 92.588 | 61.865 | 53.097 | 61.865 | 53.097 | 1.667 | 6.488 |
| 32 | 33.000 | 5.605 | 92.588 | 62.308 | 53.709 | 62.308 | 53.709 | 1.667 | 6.497 |
| 33 | 33.000 | 5.793 | 92.588 | 62.779 | 54.349 | 62.779 | 54.349 | 1.667 | 6.491 |
| 34 | 33.000 | 5.511 | 92.588 | 62.308 | 53.709 | 62.308 | 53.709 | 1.667 | 6.250 |
| 35 | 33.000 | 5.699 | 92.588 | 62.779 | 54.349 | 62.779 | 54.349 | 1.667 | 6.245 |

**Table A3.** The total $CO_2$ emissions and energy consumption in different situations.

| Number of Cases | $E_{OT}^T$ | $E_{AGV}^T$ | $E_{ARMG}^T$ $(1 \times 10^6$ kg) | $E_{QC}^T$ | $E_{Vessel}^T$ | Power Consumption $(1 \times 10^6$ kw) | Diesel Consumption $(1 \times 10^6$ L) | $CO_2$ Emission $(1 \times 10^6$ kg) | $CO_2$ Emission Sort |
|---|---|---|---|---|---|---|---|---|---|
| 1 | 59.754 | 27.666 | 418.825 | 165.667 | 491.310 | 1287.161 | 32.549 | 1163.222 | 35 |
| 2 | 59.710 | 28.611 | 422.070 | 165.667 | 491.310 | 1292.170 | 32.532 | 1167.369 | 31 |
| 3 | 59.732 | 30.029 | 429.072 | 165.667 | 491.310 | 1302.232 | 32.541 | 1175.810 | 16 |
| 4 | 59.688 | 30.974 | 432.848 | 165.667 | 491.310 | 1307.875 | 32.524 | 1180.487 | 7 |
| 5 | 59.031 | 28.139 | 422.070 | 165.667 | 491.310 | 1291.605 | 32.276 | 1166.216 | 33 |
| 6 | 58.987 | 29.084 | 425.483 | 165.667 | 491.310 | 1296.813 | 32.259 | 1170.530 | 25 |
| 7 | 59.053 | 29.556 | 429.072 | 165.667 | 491.310 | 1301.667 | 32.284 | 1174.658 | 19 |
| 8 | 59.009 | 30.502 | 432.848 | 165.667 | 491.310 | 1307.310 | 32.267 | 1179.335 | 9 |
| 9 | 59.075 | 30.974 | 436.822 | 165.667 | 491.310 | 1312.625 | 32.292 | 1183.848 | 3 |
| 10 | 58.297 | 27.666 | 422.070 | 165.667 | 491.310 | 1291.040 | 31.999 | 1165.010 | 34 |
| 11 | 58.253 | 28.611 | 425.483 | 165.667 | 491.310 | 1296.248 | 31.982 | 1169.323 | 27 |
| 12 | 58.319 | 29.084 | 429.072 | 165.667 | 491.310 | 1301.103 | 32.007 | 1173.451 | 22 |
| 13 | 58.275 | 30.029 | 432.848 | 165.667 | 491.310 | 1306.745 | 31.990 | 1178.128 | 12 |
| 14 | 57.453 | 28.139 | 425.483 | 165.667 | 491.310 | 1295.683 | 31.681 | 1168.051 | 30 |
| 15 | 57.475 | 29.556 | 432.848 | 165.667 | 491.310 | 1306.180 | 31.689 | 1176.857 | 14 |
| 16 | 57.431 | 30.502 | 436.822 | 165.667 | 491.310 | 1312.060 | 31.672 | 1181.732 | 5 |
| 17 | 57.497 | 30.974 | 441.006 | 165.667 | 491.310 | 1317.625 | 31.697 | 1186.454 | 1 |
| 18 | 56.575 | 27.666 | 425.483 | 165.667 | 491.310 | 1295.118 | 31.349 | 1166.700 | 32 |
| 19 | 56.597 | 29.084 | 432.848 | 165.667 | 491.310 | 1305.616 | 31.357 | 1175.505 | 18 |
| 20 | 56.553 | 30.029 | 436.822 | 165.667 | 491.310 | 1311.495 | 31.341 | 1180.381 | 8 |
| 21 | 56.619 | 30.502 | 441.006 | 165.667 | 491.310 | 1317.060 | 31.366 | 1185.103 | 2 |
| 22 | 55.553 | 28.139 | 429.072 | 165.667 | 491.310 | 1299.973 | 30.963 | 1169.740 | 26 |
| 23 | 55.619 | 28.611 | 432.848 | 165.667 | 491.310 | 1305.051 | 30.988 | 1174.055 | 20 |
| 24 | 55.575 | 29.556 | 436.822 | 165.667 | 491.310 | 1310.930 | 30.972 | 1178.930 | 10 |
| 25 | 54.446 | 27.666 | 429.072 | 165.667 | 491.310 | 1299.408 | 30.546 | 1168.161 | 29 |
| 26 | 54.513 | 28.139 | 432.848 | 165.667 | 491.310 | 1304.486 | 30.571 | 1172.476 | 23 |
| 27 | 54.468 | 29.084 | 436.822 | 165.667 | 491.310 | 1310.365 | 30.554 | 1177.351 | 13 |
| 28 | 54.424 | 30.029 | 441.006 | 165.667 | 491.310 | 1316.495 | 30.537 | 1182.436 | 4 |
| 29 | 53.187 | 28.611 | 436.822 | 165.667 | 491.310 | 1309.800 | 30.070 | 1175.597 | 17 |
| 30 | 53.143 | 29.556 | 441.006 | 165.667 | 491.310 | 1315.930 | 30.054 | 1180.681 | 6 |
| 31 | 51.582 | 27.666 | 432.848 | 165.667 | 491.310 | 1303.921 | 29.465 | 1169.073 | 28 |
| 32 | 51.648 | 28.139 | 436.822 | 165.667 | 491.310 | 1309.235 | 29.490 | 1173.586 | 21 |
| 33 | 51.604 | 29.084 | 441.006 | 165.667 | 491.310 | 1315.366 | 29.473 | 1178.671 | 11 |
| 34 | 49.688 | 27.666 | 436.822 | 165.667 | 491.310 | 1308.671 | 28.750 | 1171.153 | 24 |
| 35 | 49.644 | 28.611 | 441.006 | 165.667 | 491.310 | 1314.801 | 28.734 | 1176.238 | 15 |

## Appendix B. The Workflow of ARMGs in Different Operation

When the device type is selected, the relevant performance parameters are known, thus the quota power and operating speed of the device in each process can be regarded as constant, while the average running distance of the gantry will vary with the length of the block and the cooperation with other ARMGs. In general, RMG has three configurations: (a) twin cranes or non-crossing cranes, meaning that two RMGs of the same size operate on the same rail and cannot pass each other; (b) dual or crossover cranes, referring to two different sizes of RMGs (outer and inner cranes) that run on different tracks and can cross each other; or (c) triple cranes, including two twin cranes and one larger crane that move on different tracks and can pass twin cranes [57]. In this paper, we only analyze the first configuration, which means two ARMGs are arranged in one block, one is responsible for the sea side and the other is responsible for the land side. When busy, they can help each other. Therefore, compared with the traditional container terminals (rubber tire container gantry cranes are usually used as loading and unloading tools in traditional container terminals, which usually need to be transferred to other blocks when operating, and their dispatching is complicated), the movement of the ARMGs in ACTs is very simple, which only needs to move back and forth inside the block.

The storage yard is generally divided into a front yard and a rear yard. Although it is not clearly defined, generally half of the yard near the sea side is called the front yard, and the other half near the land side is the rear yard.

*Appendix B.1. Delivering*

The detailed operation of unloading the export containers transported from OTs and storing them in the front yard delivering can be seen in Figure A1. This process can be divided into four parts. Firstly, the land side RMG moves from one part of the block to the LE to unload the export container on the OT. Secondly, the land side RMG lifts the container to the relay point. Thirdly, the sea side RMG moves that container to the target yard location from the relay position. Finally, the sea side RMG comes back. Sometimes, the third and fourth steps will not be followed by the first two steps, but a centralized adjustment behind loading, known as pre-handles or pre-schedule. This article does not consider the problem of pre-handles, thus assumes that the third and fourth steps are followed by the first two steps.

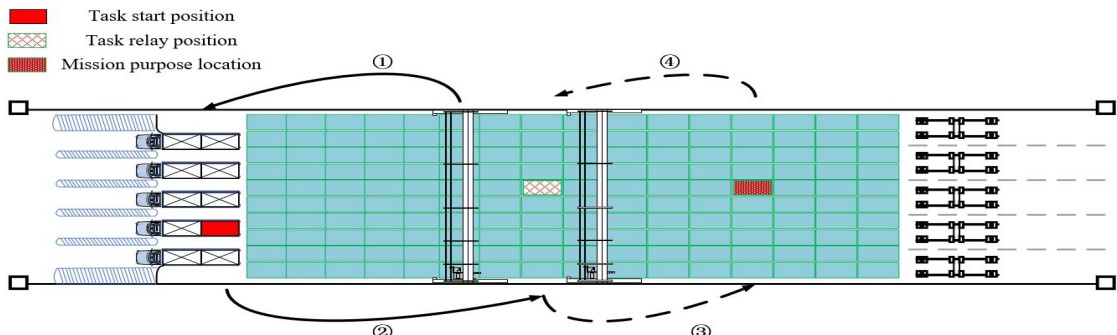

**Figure A1.** Delivering workflow of RMGs at ACTs.

We assume ARMGs move at normal speed without any interference. Therefore, for each container, the average moving distance of the ARMG can be represented as $d_{g1}^m = \frac{3}{2}L_b$.

*Appendix B.2. Loading*

Loading includes transporting the export or transferred containers stored in the front yard to the AGV or the AGV mate. The detailed operation of loading can be seen in Figure A2. This process can be divided into three parts. Firstly, the sea side ARMG moves from one part of the block to the task start position and takes out the target container by re-handling operation (not necessarily). Secondly, the sea side ARMG lifts the container to the purpose AGV or AGV mate. Finally, the sea side ARMG comes back. Because the containers arrive randomly during the process of delivering, the ARMGs generally store the containers in the order of delivering. Therefore, when loading, containers that need to be loaded on the same vessel need to be removed from any stacked containers by re-handling.

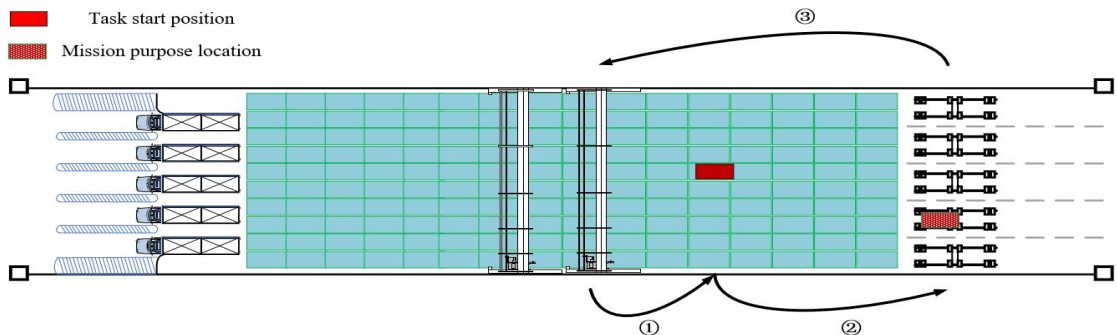

**Figure A2.** Loading workflow of ARMGs at ACTs.

Similarly, we assume ARMGs move at normal speed without any interference during the process of loading. For each container, the average moving distance of the ARMG can be represented as $d_{g1}^m = \frac{1}{2}L_b$.

*Appendix B.3. Discharging*

The detailed operation of unloading the imported or transferred containers on the AGV or AGV mate and storing them in the rear storage yard can be seen in Figure A3.

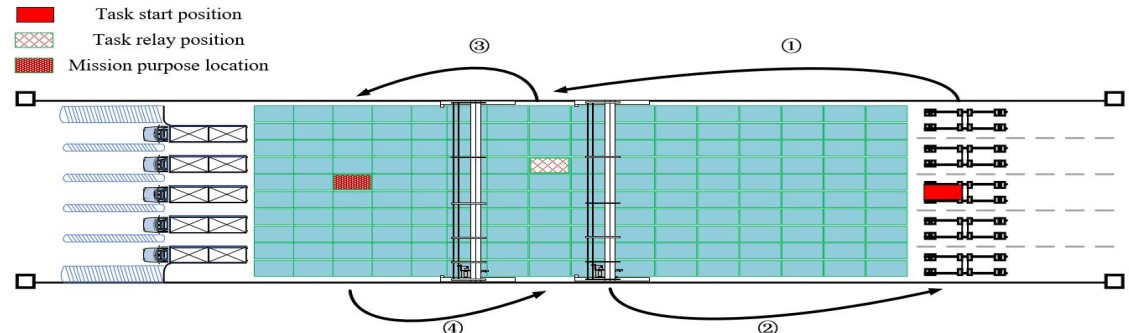

**Figure A3.** Discharging workflow of ARMGs at ACTs.

This process can be divided into four parts. Firstly, the sea side ARMG moves from one part of the block to the SE to unload the purpose container on the AGV or AGV mate. Secondly, the sea side ARMG lifts the container to the relay point. Thirdly, the land side ARMG moves that container to the target yard location from the relay position. Finally, the land side ARMG comes back. We assume ARMGs move at normal speed without any interference. Therefore, for each container, the average moving distance of the ARMG can be represented as $d_{g1}^m = \frac{3}{2}L_b$.

*Appendix B.4. Picking-Up*

The detailed operation of taking out the containers stored in the rear yard and sending them to the OTs can be seen in Figure A4.

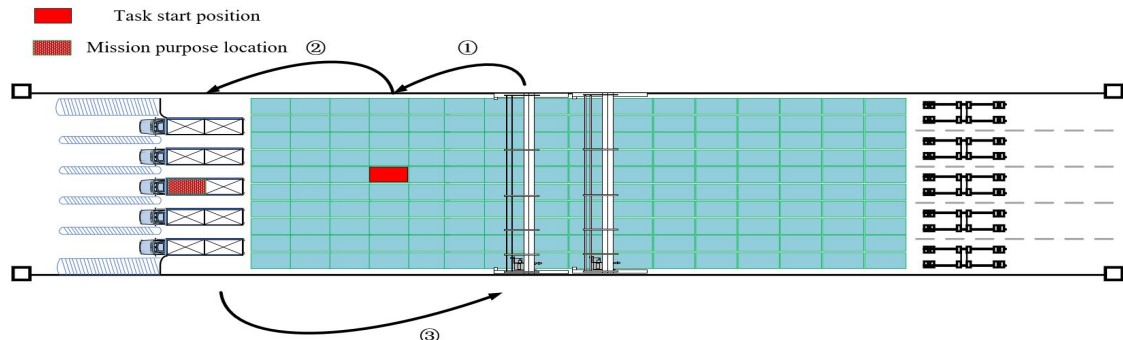

**Figure A4.** Picking-up workflow of ARMGs at ACTs.

This process can be divided into three parts. Firstly, the land side ARMG moves from one part of the block to the task start position and takes out the target container by re-handling operation (not necessarily). Secondly, the land side ARMG lifts the container to the purpose OT. Finally, the land side ARMG comes back. Similarly, the average moving distance of the ARMG can be represented as $d_{g1}^m = \frac{1}{2}L_b$. Because the ARMGs generally store the containers in the order of discharging, the OTs arrive randomly during the process of picking-up. Therefore, when picking-up, containers that need to be picked need to be removed from any stacked containers by re-handling.

**Appendix C. The Specific Process of QCs in ACTs**

The specific loading operation diagrams of the main trolley and portal trolley are shown as the solid line and the dotted line of Figure A5, respectively.

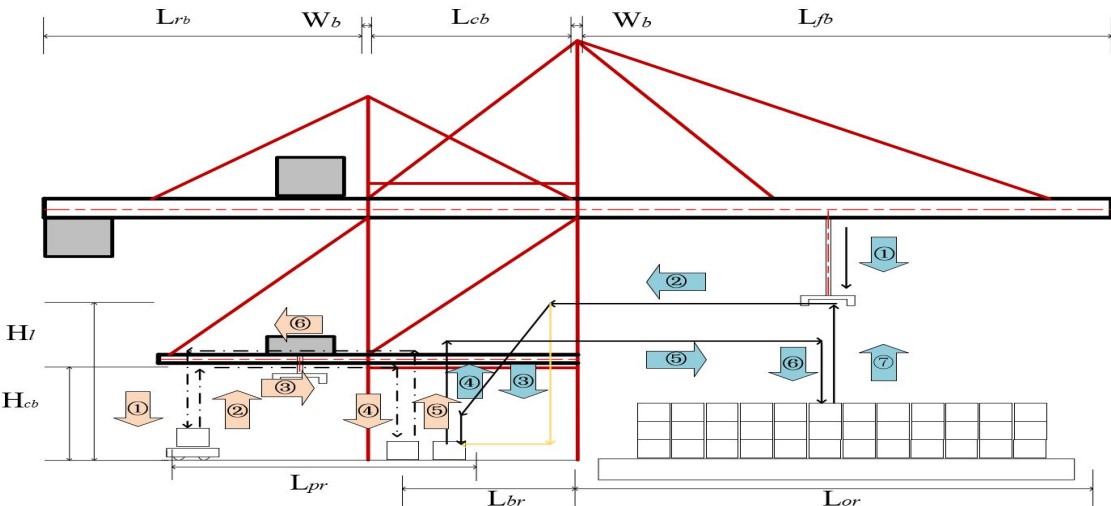

**Figure A5.** The specific operation diagram of the main trolley and portal trolley.

Assume that the default position of the main trolley starts at the highest position of the sea side rail of the main trolley. After the loading operation starts, it drops from the default position to the lifting height, then moves to the back rail through traverse travel, and reaches to the top of the ground container to grab the container through oblique motion (moving down and to the land side simultaneously, as shown by the solid black line in the figure). After the spreader is stabilized, it hoists to the second lifting height through the original road, which is a little higher than the rail of the portal trolley, and transports to the top of the vessel through traverse travel, and then lowers to the vessel to place the container. After the container is stabilized, the spreader is opened and rises to the lifting height, and waits for next operation. Therefore, a circular path diagram of the main trolley is shown by the blue arrow ②③④⑤⑥⑦ in the figure during loading. The discharging process is reversed. The operation process of the portal trolley is similar. It begins to lower from the portal rail to grab the container above the AGV. After the spreader grabs the container, it rises through the original road. It transports to the top of the transfer platform through traverse travel and then descends. After the container is stabilized, the spreader is opened and rises to portal rail. Finally, it comes back to the top of the initial point through horizontal transportation and waits for next operation. Thus, a circular path diagram of the portal trolley is shown by the orange arrow ①②③④⑤⑥ in the figure during loading. The discharging process is reversed, too. Thus, the operation process of portal trolley can be simplified to two up and down movements without load, two up and down movements with load, one transverse movement without load, and one transverse movement with load. The operation process of main trolley is similar. (The third step is calculated according to the path of the yellow solid line in modeling.)

According to Figure A5, for portal trolley, the average distance of one horizontal transport can be regarded as half the length of the portal rail. The average distance of one vertical transport can be regarded as the height of the portal rail minus the height of one container. For main trolley, the vertical transport distance with/without load on the sea side can be regarded as the height of the portal rail/lifting (above rail) minus half of the container stack height on the vessel (if the stock height is lower than the land height, it is considered negative); the vertical transport distance with/without load on the land side can be regarded as the height of the portal rail/lifting (above rail) minus the height of one container; and the one horizontal transport with/without load distance can be regarded as half of the width of the vessel plus the back reach, thus the distance in the operation of discharging. For the gantry, it only does horizontal movement when container tasks at assigned bay are completed. The average power consumption of the gantry in one operation can be regarded as the total energy consumption divided by the number of handling containers. The average moving distance varies

depending on the assignment task of the QC. Therefore, the energy consumption of QCs in one operation can be expressed as follows.

$$
\begin{aligned}
\overline{E_{QC}} = {} & (L_{br} + \frac{1}{2}W_v) * (\frac{P_m^{tl}}{v_m^{tl}} + \frac{P_m^{tu}}{v_m^{tu}}) + (2H_{pr} - \frac{1}{2}H_c - h_c) * \frac{P_m^{hl}}{v_m^{hl}} + (2H_l - \frac{1}{2}H_c - h_c) * \frac{P_m^{hu}}{v_m^{hu}} \\
& + \frac{1}{2}L_{pr} * (\frac{P_p^{tl}}{v_p^{tl}} + \frac{P_p^{tu}}{v_p^{tu}}) + 2(H_{pr} - h_c) * (\frac{P_p^{hl}}{v_p^{hl}} + \frac{P_p^{hu}}{v_p^{hu}}) + \frac{d_g}{v_g} * P_g / n_c
\end{aligned}
\tag{A1}
$$

where $W_v$ is width of the vessel; $L_{pr}$ is length of the portal rail; $L_{br}$ is back reach; $H_{pr}$ is height of the portal rail; $h_c$ is height of one container; $H_l$ is height of lifting (above rail); $H_c$ is stack height of containers on the vessel; and $n_c$ is the number of containers handled by the QC.

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
