# Peer review of "Analysis and Design of Typical Automated Container Terminals Layout Considering Carbon Emissions"

_sustainability, doi:10.3390/su11102957_

Reviewer 1 Report

The authors have significantly improved the quality of their paper. However they still need to address a few things, ensure that a native proofreader improves the paper, and importantly reduce the size of the manuscript as it is currently too long.

A few specific suggestions for improvements below:
In your introduction section you are stating some important statistical data but you are not providing references for these numbers. For example the 60% in line 16 (presumably from UNCTAD), the start of construction of ACTs in Asia and Europe and so on.

Line 33 refrain from suggesting that construction costs are high. Provide a number instead.

Table 1 you need to provide references for the numbers. Presumably a report from each container terminal, or data from IAPH?
Improve the presentation of Figure 2 as it is currently poor. While it is interesting, this is only one port so please have a short discussion on the transferability of the findings in Figure 2.

Section 2.2 and 2.3 need to be rewritten. The focus near ports is not the carbon emissions (that is a generic concern of shipping), but perhaps more improtantly local pollutants (SOX, NOx, PM). Also, line 225 you are using first person tense. This is unacceptable. There are several small errors like this one, so I would strongly encourage that you use a native proof reader before resubmitting.

For example, line 210 you write "In generally", it should be "In general". etc.

Figures 3-7 are taken from somewhere or the authors produced it?

Figures 11 and 12 could be improved (small fonts, reduce text).

Tables 9 and 11 are necessary? you have to discuss these results a bit more.

Are the Appendices really necesary? I am not sure if these are adding value (also many typos for example peocess instead of process in the Title! of Appendix B    

Author Response

Point 1: The authors have significantly improved the quality of their paper. However they still need to address a few things, ensure that a native proofreader improves the paper, and importantly reduce the size of the manuscript as it is currently too long. 

Response 1: Thanks to the reviewer for its comments. The grammatical aspects of the article have been corrected and marked in blue. The size of the manuscript has been shortened. Except for Abbreviations, Appendix, References, the content of this article in the latest version is less than 29 pages.

Point 2: In your introduction section you are stating some important statistical data but you are not providing references for these numbers. For example the 60% in line 16 (presumably from UNCTAD), the start of construction of ACTs in Asia and Europe and so on.

Response 2: Thanks to reviewer for pointing out the lack of references of data in our articles. In the latest version, we have updated the relevant data and marked it in red.

Point 3: Line 33 refrain from suggesting that construction costs are high. Provide a number instead.

Response 3: Thanks to the reviewer for pointing out that the lack of data support in the statements in our article. In the latest version, we used the cost of TRAPAC TERMINAL PROGRAM to show the construction costs are high. The TRAPAC TERMINAL PROGRAM provided wharves, automated backlands, rail facilities, buildings, and gates for the Port of Los Angeles’ first automated container terminal at Berths 136-147 and delivered the first automated terminal on the West Coast of USA.

Point 4: Table 1 you need to provide references for the numbers. Presumably a report from each container terminal, or data from IAPH?

Response 4: Thanks to reviewer for pointing out the lack of references of Table 1 in our articles. In the latest version, we have updated the reference for Table 1 and marked it in red.

Point 5: Improve the presentation of Figure 2 as it is currently poor. While it is interesting, this is only one port so please have a short discussion on the transferability of the findings in Figure 2.

Response 5: Thanks to the reviewer for pointing out the shortcomings of Figure 2.

Figure 2 shows the total maritime industry-related CO2 emissions in the port of Los Angeles ( POLA) from 2005 to 2017. The POLA and the port of Long Beach have established the San Pedro Bay Ports Clean Air Action Plan (CAAP) to reduce pollution from their production since 2005. As the biggest container terminal in the USA, the POLA has published a full report each year called “Air Emissions Inventory” since the beginning of CAAP. Based on the data from the report, Figure 2 shows the trend in the CO2 equivalent emissions over 2005–2017. It can be seen from the figure that since the implementation of CCAP in 2005, carbon emissions of POLA have shown a significant downward trend, while in recent years, there have been some fluctuations due to the increase in the number of port containers.

We want to use picture 2 to show that although with the implementation of measures such as "oil to electricity" of terminal equipment, restriction of ship route and speed, emission reduction has achieved certain results, more efforts still need to be tried to reduce carbon emissions, such as optimizing the layout like our article.

Point 6: Section 2.2 and 2.3 need to be rewritten. The focus near ports is not the carbon emissions (that is a generic concern of shipping), but perhaps more improtantly local pollutants (SOX, NOx, PM). Also, line 225 you are using first person tense. This is unacceptable. There are several small errors like this one, so I would strongly encourage that you use a native proof reader before resubmitting.

For example, line 210 you write "In generally", it should be "In general". etc.

Response 6: Thanks to the reviewer for its comments. The grammatical aspects of the article have been corrected and marked in blue. In terms of the focus near ports, we do agree pollutants like SOX, NOx, PM is very important, while as we have discussed in section 1, carbon emission is closely related to global warming and CO2 is something that governments and port operators are eager to reduce. In fact, the calculation formula for pollutants is the same, but the parameters are different (CO2 emissions are significantly larger than other pollutants), so this paper only considers CO2 emissions.

Point 7: Figures 3-7 are taken from somewhere or the authors produced it?

Response 7: Thanks to the reviewer for its comments. Figure 3-7 are produced by ourselves based on the software Visio.

Point 8: Figures 11 and 12 could be improved (small fonts, reduce text).

Response 8: Thanks to the reviewer for its comments. Figures 11 and 12 are improved in the enclosed manuscript according to reviewer’s suggestion.

Point 9: Tables 9 and 11 are necessary? you have to discuss these results a bit more.

Response 9: Thanks to the reviewer for its comments. Table 9 is the appropriate layout combination table calculated for the case Automated container terminal (ACT), which contains the width of each area in each situation, corresponding to the description of the ACT layout in section 3.1. Considering that there are three decision variables affecting the layout results at the same time, in section 5.3, the analysis of the single region and the correlation analysis of decision variable is carried out successively through the data from Table 9 and Table 10. Table 10 shows the average energy consumption of each equipment during cycle operation process in different situations. Table 11 shows the total carbon emissions and energy consumption of each equipment for each suitable layout case calculated for the case ACT. The data from the two tables can be compared to analyse the average and total carbon emissions of each device. This is also shown in Figure 24 and the results are discussed. Considering that the article is too long, these tables are moved to the appendix in the latest version.

Point 10: Are the Appendices really necesary? I am not sure if these are adding value (also many typos for example peocess instead of process in the Title! of Appendix B   

Response 10: Thanks to the reviewer for its comments. The grammatical aspects of the article have been corrected. The appendix B and C details the workflow of ARMGs in different operation and the specific process of QCs in ACTs and explains the parameters used in my calculation of the device cycle operation. If the appendix is removed, the reader may question the parameters I brought in the calculation. So in the submitted manuscript, the appendix has been retained.

Reviewer 2 Report

A very interesting and extensive article. Maybe even too wide. However, an in-depth analysis of the literature was made, a new mathematical model was applied and its functioning was proven. The logical arrangement of the article finds a reference to sustainable development, which accomplishes the purpose of the article placed in the introduction.

It is important to read the whole thoroughly and make language and punctuation corrections, as well as consider the form and clarity of diagrams, charts, figures and tables, etc.

Author Response

Point 1: A very interesting and extensive article. Maybe even too wide. However, an in-depth analysis of the literature was made, a new mathematical model was applied and its functioning was proven. The logical arrangement of the article finds a reference to sustainable development, which accomplishes the purpose of the article placed in the introduction.

Response 1: Thanks to the reviewer for its comments. The grammatical aspects of the article have been corrected and marked in blue. The size of the manuscript has been shortened. Except for Abbreviations, Appendix, References, the content of this article in the latest version is less than 29 pages.

Point 2: It is important to read the whole thoroughly and make language and punctuation corrections, as well as consider the form and clarity of diagrams, charts, figures and tables, etc.

Response 2: Thanks to the reviewer for its comments. According to the reviewer's suggestion, we repeatedly checked the content of this article and improved some of the clarity of diagrams, charts, figures and tables in the latest version. The grammatical aspects of the article have been corrected and marked in blue.

Reviewer 3 Report

The authors have put a lot of work into improving the article - now its scientific value is definitely higher.
Now the only downside is the length of the article - it can disturb readers. I understand, however, that this is dependent on the problem and research.

Author Response

Point 1: The authors have put a lot of work into improving the article - now its scientific value is definitely higher.

Now the only downside is the length of the article - it can disturb readers. I understand, however, that this is dependent on the problem and research.

Response 1: Thanks to the reviewer for its comments. In order to clearly clarify the content of this article, we have retained a lot of content in the latest version and also made appropriate deletions. Except for Abbreviations, Appendix, References, the content of this article in the latest version is less than 29 pages.

Round  2

Reviewer 1 Report

    Authors have addressed my comments

This manuscript is a resubmission of an earlier submission. The following is a list of the peer review reports and author responses from that submission.

Round  1

Reviewer 1 Report

The paper is dealing with an interesting subject however there are many flaws in the analysis, and certain things need to be significantly revised.
The title is misleading, you should not be using the term optimal for the layout of your terminal. In your paper you are not optimizing, you are just presenting a simple emissions model for the various functions taking place in the terminal. In section 3 you are simply using some closed loop formulas to determine some of the key variables (e.g. number of lanes etc).
The carbon emissions model section is unnecessarily long.
In particular in section 4.5 on vessels, the authors are using an outdated paper of Psaraftis and Kontovas to estimate emissions at berth. That paper is a seminal work when it comes to emissions from shipping, but with a focus on cruise emissions.

For near port emissions, I would recommend that you use instead the methodology of Zis who is focusing solely on near and at ports emissions. See below:

Zis, T., North, R. J., Angeloudis, P., Ochieng, W. Y., & Bell, M. G. H. (2014). Evaluation of cold ironing and speed reduction policies to reduce ship emissions near and at ports. Maritime Economics & Logistics, 16(4), 371-398.

Zis, T. P. (2019). Prospects of cold ironing as an emissions reduction option. Transportation Research Part A: Policy and Practice, 119, 82-95.

Also you need to provide some data. How many ships are you expecting, how is the layout of the terminal going to affect ship operations (for example a layout that minimizes movemetns atthe yard may not be optimal for serving vessels).

Section 5 needs to be condensed and presented more clearly. It currently is very confusing and your graphs are not helping interpret your results.
Section 6; It is not clear to me how you ended up with relationship 47.

In general, I think that the topic you have selected is interesting, but you need to narrow it down and since you are not optimizing, rewrite the paper and simply compare a few different designs of terminal layouts. Use a simulation software to deal with queueing phenomena.

Reviewer 2 Report

The article is interesting, pragmatic. He contributes to the issue of the Sustainability magazine.
Literature review sets a research gap, which the authors want to fill. The review of lietratura is largely based on the output of Chinese authors, which is not bad, but maybe it is worth looking at more of the world literature, including studies, research results, the world case study (a'la comparison), etc.
The research is conducted in a clear and comprehensible manner, which is an advantage of the article. The drawings allow you to understand the course of thought in research even more.
Conclusions and discussion are sufficient.
I also have some comments:
1. Abstract has no standard, generally accepted structure. It lacks, among others the purpose of the article and the description of the research method.

2. I think that putting the description in the formulas at the beginning of the article is not good. It's better to put it at the end of the article.

Reviewer 3 Report

1. The article is interesting but of low originality. This means that the problem is already widely described in the world. The authors did not make a thorough analysis of world literature - American and European. The indicated world (foreign) positions of literature are modest and not analyzed. Hence, probably the low originality of the article.

2. Taking into account the nature of the journal, the indicated article raises too many threads in terms of sustainability and sustainable development. Unfortunately, these threads are not connected by a common research ground - in this case, a common research ground for all topics to be discussed should be the sustainability or specific aspect, e.g. the impact of carbon dioxide emissions on human life, or ecological efficiency, etc.

3. The optimization of container terminals and the problem of decarbonisation should be treated as a separate thread in the context of sustainability, that is, in relation to particular aspects of sustainability or the impact of optymimalization on the level of sustainability. The analysis and methods used in the article show the relationship between the optimization of container terminals and carbon dioxide emission but do not indicate the significance of these phenomena.

4. The carbon emission calculator is a common tool used in the world and in the generally available addon. If a specific model for the described problem was proposed, why were there no effects and differences between what we know and use (Carbon Carbon Emulator) and the new model proposed by the authors? Relationships have not been demonstrated and again not referred to sustainability.

5. A completely separate thread raised by the authors without connection to the previous considerations is the problem of energy consumption. I understand that the authors wanted to show the impact of the applied solutions on the level of energy intensity, but it still does not follow from the article why it is happening and it is a sustainable problem. Combining this with considerations about decarbonisation is not clear.

6. The authors mixed up the threads. Therefore, the article does not meet the set goal. I suggest dividing the article into three parts - the optimization model of container terminals with reference to the problem of carbon emission in the light of the sustainable development of ports; energy consumption of container terminal operations in the ecological aspect of sustainability - or similar, e.g. in the context of the model; problem of calculating energy consumption.

7. The research methodology should be explained. The abbreviations used should be clarified. The correlation should be presented in tabular form. It is necessary to divide the research into various threads - and therefore different articles.

8. In its present form, the article is not a scientific problem and is not original, but above all it does not refer to the problem of sustainable ports - which is the main demand of the jurnal.